# Hydroxychloroquine/chloroquine for the treatment of hospitalized patients with COVID-19: An individual participant data meta-analysis

Leon Di Stefano[1], Elizabeth L. Ogburn[1], Malathi Ram[2,3], Daniel O. Scharfstein[4], Tianjing Li[5], Preeti Khanal[3], Sheriza N. Baksh[6], Nichol McBee[3], Joshua Gruber[3], Marianne R. Gildea[3¤], Megan R. Clark[3], Neil A. Goldenberg[7,8,9], Yussef Bennani[10,11], Samuel M. Brown[12,13], Whitney R. Buckel[14], Meredith E. Clement[10,11], Mark J. Mulligan[15,16], Jane A. O'Halloran[17], Adriana M. Rauseo[17], Wesley H. Self[18], Matthew W. Semler[19], Todd Seto[20], Jason E. Stout[21], Robert J. Ulrich[15], Jennifer Victory[22], Barbara E. Bierer[23,24], Daniel F. Hanley[3], Daniel Freilich[25]*, on behalf of the Pandemic Response COVID-19 Research Collaboration Platform for HCQ/CQ Pooled Analyses[¶]

1 Department of Biostatistics, Johns Hopkins Bloomberg School of Public Health, Baltimore, Maryland, United States of America, 2 Department of International Health, Johns Hopkins Bloomberg School of Public Health, Baltimore, Maryland, United States of America, 3 Division of Brain Injury Outcomes, Johns Hopkins School of Medicine, Baltimore, Maryland, United States of America, 4 Division of Biostatistics, Department of Population Health Sciences, University of Utah School of Medicine, Salt Lake City, Utah, United States of America, 5 University of Colorado Denver, Anschutz Medical Campus, Denver, Colorado, United States of America, 6 Johns Hopkins Bloomberg School of Public Health, Baltimore, Maryland, United States of America, 7 Department of Pediatrics, Johns Hopkins School of Medicine, Baltimore, Maryland, United States of America, 8 Department of Medicine, Johns Hopkins School of Medicine, Baltimore, Maryland, United States of America, 9 Johns Hopkins All Children's Institute for Clinical and Translational Research, Johns Hopkins All Children's Hospital, St. Petersburg, Florida, United States of America, 10 Louisiana State University Health Sciences Center, New Orleans, Louisiana, United States of America, 11 University Medical Center, New Orleans, New Orleans, Louisiana, United States of America, 12 Division of Pulmonary and Critical Care Medicine, Intermountain Medical Center, Murray, Utah, United States of America, 13 University of Utah, Salt Lake City, Utah, United States of America, 14 Pharmacy Services, Intermountain Healthcare, Murray, Utah, United States of America, 15 Department of Medicine, Division of Infectious Diseases and Immunology, New York University Grossman School of Medicine, New York, New York, United States of America, 16 Vaccine Center, New York University Grossman School of Medicine, New York, New York, United States of America, 17 Department of Medicine, Washington University School of Medicine, Saint Louis, Missouri, United States of America, 18 Department of Emergency Medicine, Vanderbilt University Medical Center, Nashville, Tennessee, United States of America, 19 Division of Allergy, Pulmonary and Critical Care Medicine, Department of Medicine, Vanderbilt University Medical Center, Nashville, Tennessee, United States of America, 20 Department of Medicine, University of Hawaii John A. Burns School of Medicine, Honolulu, Hawaii, United States of America, 21 Division of Infectious Diseases and International Health, Duke University Medical Center, Durham, North Carolina, United States of America, 22 Bassett Research Institute, Bassett Medical Center, Cooperstown, New York, United States of America, 23 Department of Medicine, Brigham and Women's Hospital, Boston, Massachusetts, United States of America, 24 Harvard Medical School, Boston, Massachusetts, United States of America, 25 Department of Internal Medicine, Division of Infectious Diseases, Bassett Medical Center, Cooperstown, New York, United States of America

¤ Current address: FHI 360, Durham, North Carolina, United States of America
¶ Membership of the Pandemic Response COVID-19 Research Collaboration Platform for HCQ/CQ Pooled Analyses is provided in the Acknowledgments.
* daniel.freilich@bassett.org



**Data Availability Statement:** The ORCHID trial data underlying the results presented in the study are available from the National Heart, Lung, and Blood Institute Biologic Specimen and Data

## Abstract

Repository Information Coordinating Center (https://biolincc.nhlbi.nih.gov/; accession number HLB02372021a). The data for the other studies presented are available from Vivli (https://www.vivli.org): COVID MED, https://doi.org/10.25934/00006535; HAHPS, https://doi.org/10.25934/00006626; NCT04335552, https://doi.org/10.25934/00006861; NCT04344444, https://doi.org/10.25934/00006865; OAHU-COVID19, https://doi.org/10.25934/00006595; TEACH, https://doi.org/10.25934/00006627; and WU352, https://doi.org/10.25934/00006713. Policies for accessing these third-party datasets vary somewhat by study and repository, but requests must be approved and require a signed agreement.

**Funding:** This work was supported by the National Institutes of Health (NIH) National Center for Advancing Translational Sciences (grant U24 TR001609, awarded to D.F.H.). L.D.S. is supported by an American Australian Association Sir Keith Murdoch Scholarship. E.L.O. is supported by the Johns Hopkins Bloomberg School of Public Health. S.M.B. and W.R.B. were funded by the Intermountain Research and Medical Foundation, Intermountain Heart and Lung Foundation, and Intermountain Office of Research for the HAHPS study. J.A.O. and the WU352 study were funded by an NIH Clinical and Translational Science Award (grant UL1 TR002345) through the Washington University Institute of Clinical and Translational Sciences. W.H.S. and the ORCHID trial were supported by the National Heart, Lung, and Blood Institute (grants 3U01 HL123009-06S1, U01 HL123009, U01 HL122998, U01 HL123018, U01 HL123023, U01 HL123008, U01 HL123031, U01 HL123004, U01 HL123027, U01 HL123010, U01 HL123033, U01 HL122989, U01 HL123022, and U01 HL123020). Sandoz, a Novartis division, supplied the hydroxychloroquine and placebo used in the ORCHID trial. J.E.S. was funded by Duke University School of Medicine for the NCT04335552 study. J.V. and D.F. are supported by the Bassett Research Institute. B.E.B. is supported by the National Center for Advancing Translational Sciences (grant UL1 TR002541) and Brigham and Women's Hospital. The funders had no role in study design, data collection and analysis, decision to publish, or preparation of the manuscript.

**Competing interests:** The authors have read the journal's policy and have the following competing interests: S.N.B., N.M., M.R.C., and D.F.H. report receiving research funding from the Department of Defense for clinical trials of convalescent plasma for COVID-19 outside the submitted work. N.A.G. reports receiving salary support from the National

## Background

Results from observational studies and randomized clinical trials (RCTs) have led to the consensus that hydroxychloroquine (HCQ) and chloroquine (CQ) are not effective for COVID-19 prevention or treatment. Pooling individual participant data, including unanalyzed data from trials terminated early, enables more detailed investigation of the efficacy and safety of HCQ/CQ among subgroups of hospitalized patients.

## Methods

We searched ClinicalTrials.gov in May and June 2020 for US-based RCTs evaluating HCQ/CQ in hospitalized COVID-19 patients in which the outcomes defined in this study were recorded or could be extrapolated. The primary outcome was a 7-point ordinal scale measured between day 28 and 35 post enrollment; comparisons used proportional odds ratios. Harmonized de-identified data were collected via a common template spreadsheet sent to each principal investigator. The data were analyzed by fitting a prespecified Bayesian ordinal regression model and standardizing the resulting predictions.

## Results

Eight of 19 trials met eligibility criteria and agreed to participate. Patient-level data were available from 770 participants (412 HCQ/CQ vs 358 control). Baseline characteristics were similar between groups. We did not find evidence of a difference in COVID-19 ordinal scores between days 28 and 35 post-enrollment in the pooled patient population (odds ratio, 0.97; 95% credible interval, 0.76–1.24; higher favors HCQ/CQ), and found no convincing evidence of meaningful treatment effect heterogeneity among prespecified subgroups. Adverse event and serious adverse event rates were numerically higher with HCQ/CQ vs control (0.39 vs 0.29 and 0.13 vs 0.09 per patient, respectively).

## Conclusions

The findings of this individual participant data meta-analysis reinforce those of individual RCTs that HCQ/CQ is not efficacious for treatment of COVID-19 in hospitalized patients.

## Introduction

During the COVID-19 pandemic Delta and Omicron variant surges, US daily deaths again reached 1,000–2,000, reinforcing the need for effective therapeutics. Early in the pandemic, hydroxychloroquine (HCQ) and chloroquine (CQ) received a Food and Drug Administration (FDA) Emergency Use Authorization (EUA) for treatment of hospitalized COVID-19 patients, and the drugs were administered empirically and recommended in some guidelines [1]. Supportive efficacy data relied on inconsistent results from preclinical studies [2] and small uncontrolled trials [3]. Based in part on these early data, an estimated 42% of hospitalized COVID-19 patients in the US received HCQ in March 2020 [4].

Subsequently, most retrospective-observational studies of HCQ/CQ in hospitalized COVID-19 patients found no evidence of benefit, and possibly higher mortality, with concerns about toxicities (e.g., QTc prolongation) [5–15]. Results from at least 5 randomized clinical trials (RCTs) became available in spring/summer 2020; all showed no evidence of benefit, and

Institutes of Health (NIH) National Center for Advancing Translational Sciences via a Johns Hopkins Clinical and Translational Science Award outside the submitted work. Y.B. reports being a site investigator for Janssen outside the submitted work. S.M.B. reports service as chair of a data and safety monitoring board for a Hamilton clinical trial in respiratory failure; fees paid to Intermountain Healthcare from Faron Pharmaceuticals and Sedana Pharmaceuticals for steering committee service for a clinical trial in acute respiratory distress syndrome; research grants to Intermountain Healthcare from Janssen, NIH, Centers for Disease Control and Prevention, and Department of Defense; and royalties from Oxford University Press and Brigham Young University, outside the submitted work. M.E.C. reports service on a Roche advisory board and as a site investigator for Janssen outside the submitted work. This does not alter our adherence to PLOS ONE policies on sharing data and materials.

most showed adverse safety signals [14, 16–19]. In light of these results, most principal investigators discontinued enrollment in HCQ/CQ arms of their trials. Consequently, adequate power to reach robust conclusions regarding efficacy and safety of HCQ/CQ was no longer attainable for many incomplete trials; moreover, effect estimates in published trials were accompanied by wide confidence intervals. Nevertheless, at least 11 additional RCTs published later in the pandemic found similar results [20–30].

The goals of this study were to ensure utilization of data from unpublished RCTs evaluating HCQ/CQ by combining them with published data and to synthesize evidence on HCQ/CQ efficacy and safety in hospitalized COVID-19 patients, overall and in subpopulations of interest, by conducting an individual participant data (IPD) meta-analysis. This study design involves pooling subject-level data from multiple studies and possesses many advantages over both individual randomized trials and traditional aggregate-data meta-analyses. Individual trials are usually designed to detect overall effects; the increased sample size in an IPD meta-analysis can enable more precise estimation of subgroup effects [31]. A more diverse sample in a pooled analysis can also improve external validity over individual trials [32]. Compared with aggregate data meta-analyses, IPD meta-analyses are less vulnerable to the ecological fallacy, allow for consistent analytic choices within each study, and enable researchers to consider subgroup effects that were not considered in the original studies [33].

## Methods

This project was approved by the Johns Hopkins Medicine institutional review board, with individual studies approved by their local ethics boards as deemed necessary.

### Trials selection summary

The Trial Innovation Network, a National Center for Advancing Translational Sciences initiative to increase efficiency and innovation in clinical research, partnered with the COVID-19 Collaboration Platform to promote coordination among research groups running similar trials. The team contacted principal investigators of COVID-19 RCTs registered on Clinical-Trials.gov starting on April 30, 2020, and encouraged uploading study protocols to the COVID-19 Collaboration Platform (CovidCP) repository (http://covidcp.org). The platform, initiated with the goal of sharing protocols to facilitate collaboration, aims to combine data or aggregate evidence from similar studies to increase efficiency and precision.

One respondent (Bassett) had independently initiated a collaboration registry effort and performed systematic searches of ClinicalTrials.gov on May 9, 2020, and May 21, 2020, using search terms "COVID-19" and "hydroxychloroquine OR chloroquine," and study status of "recruiting."

Trials from the Bassett search and CovidCP repository were aggregated. Additional outreach by investigators occurred in June 2020 to studies located at Clinical and Translational Science Awards Program institutions (S3 Appendix). This combined list was the primary driver for study selection, with augmentation and refinement by additional systematic ClinicalTrials.gov searches through June 2, 2020.

US-based RCTs of HCQ/CQ to treat patients with SARS-CoV-2 infection were eligible for inclusion if patient informed consent and/or individual study institutional review board approval allowed data sharing; study institutions signed a data use agreement for the present study; the outcomes as defined in this study were recorded or could be extrapolated; and trialists agreed to participate. We excluded trials in non-hospitalized patients, trials not registered on ClinicalTrials.gov, trials without enrollment, and international trials to avoid data sharing

regulatory delays. We decided to focus on inpatient studies. No individual-level exclusion criteria were imposed beyond those employed by each study.

## Data collection and harmonization

A common data harmonization tool, including a data dictionary with definitions and encodings of variables, example data, and deidentification functions for dates and ages consistent with Health Insurance Portability and Accountability Act requirements, was used by 7 trial teams to create data sets that were uploaded to the data repository Vivli (https://www.vivli.org) and then downloaded by the CovidCP team. ORCHID trial data were downloaded from the National Heart, Lung, and Blood Institute Biologic Specimen and Data Repository Information Coordinating Center (BioLINCC) and manually harmonized by the CovidCP team. Queries about missing, unusual, or inconsistent data were resolved via direct contact with studies' principal investigators and, in some cases, manual chart review.

## Outcomes

The primary outcome was clinical improvement measured on a 7-point ordinal scale with levels (1) death; (2) hospitalized, on mechanical ventilation or extracorporeal membrane oxygenation (ECMO); (3) hospitalized, on non-invasive ventilation (BiPAP/CPAP and/or high-flow oxygen); (4) hospitalized, requiring oxygen; (5) hospitalized, not requiring oxygen; (6) not hospitalized, with limitation; and (7) not hospitalized, without limitations. This scale is relatively coarse compared to others in use (for example, the 11-point World Health Organization [WHO] scale [34]), and was chosen to make the data easier to harmonize. We prespecified an outcome window of day 28–30 post-enrollment, which was broadened to day 28–35 after data collection due to missingness (see Results). Differences in the primary outcome were assessed using proportional odds ratios.

Secondary outcomes included hospital length of stay, need for mechanical ventilation, and 28-35-day mortality. Safety outcomes included rates of overall adverse events (AEs) and serious adverse events (SAEs), and rates of specific AEs and SAEs of interest: elevated liver function tests (LFTs), QTc prolongation, and arrhythmias. Due to practical constraints, we did not attempt to synchronize adverse event definitions across the included studies.

## Baseline and post-baseline variables

From each trial, baseline variables included treatment assignment, age (5-year interval bins), sex, race and ethnicity, body mass index (BMI), symptom duration, mechanical ventilation status, ordinal score, and comorbidities (cerebrovascular disease, myocardial infarction, congestive heart failure, dementia, chronic obstructive pulmonary disease, asthma, hypertension, tumor, liver disease, diabetes, smoking, and vaping), as well as post-baseline (enrollment through day 28) azithromycin and corticosteroid use (S1 Table).

## Statistical analysis

The primary outcome was analyzed in two ways. First, we fit a proportional odds model with treatment indicator as the sole covariate using the "polr" command in R (version 4.0.4). Second, we fit a Bayesian proportional odds regression model including a main effect for treatment; fixed effects for sex and baseline ordinal scale (disease severity); splines of age, BMI, and a number of baseline comorbidities; and random effects for baseline ordinal scale and study. The fixed and random effects were also interacted with treatment. All fixed regression coefficients were given uniform priors. Random effects were modeled as independent, with standard

deviations given independent half-t priors with 3 degrees of freedom and scale 10. The model was fit using the R package "brms" (version 2.15). Missing baseline covariates were imputed using multiple imputation by chained equations, as implemented in the R package "mice" (version 3.12); missing outcomes were treated as missing at random conditional on the included covariates. Inferences were based on fitting the model separately to each imputed data set, then pooling posterior draws across the imputations. The model was used to obtain standardized estimates of the overall treatment effect, where standardization was with respect to the empirical distribution of the baseline covariates in the pooled study population. Relative to the first approach, we employed the second approach to leverage covariates to produce more stable and accurate inferences, particularly in small subgroups.

The following subgroup analyses were prespecified: study; sex; age ($\leq$29, 30–49, 50–69, 70–79, 80+ years); disease severity as measured by baseline ordinal score (2, 3, 4, 5); and BMI ($\leq$20, 20–25, 25–30, 30–35, >35). Prespecified subgroup analyses based on Charlson score were replaced with a simple baseline comorbidities count (0, 1, 2, 3, $\geq$4) due to systematic missingness in component variables. Subgroup analyses were conducted using the two approaches discussed above. For the first, a proportional odds model was fit separately within each subgroup. For the second, the Bayesian regression model above was used to obtain standardized subgroup estimates, where standardization was with respect to the empirical distribution of covariates within subgroups. We conducted two post-hoc subgroup analyses. The first used quintiles of a baseline risk score given by the expected linear predictor for each study participant under the control condition, as per recommendations from Kent et al. [35]. The second was based on the time between symptom onset and enrollment (0–4 days, 5–7 days, $\geq$8 days; groups based on approximate tertiles).

All-cause 28-35-day mortality was analyzed using the same approaches. Other secondary and safety outcomes were analyzed descriptively.

To assess the sensitivity of our conclusions to the choice of model and outcome window, we (1) repeated our analysis with weakly informative priors; (2) fit an expanded model including terms for assignment to an azithromycin arm and days between symptom onset and enrollment; (3) fit a version of the main model with no treatment interactions; (4) expanded and contracted outcome windows to 28–40 and 28–30 days, respectively; and (5) re-ran our analysis with a model fit only to ORCHID's data set, the largest of the 8 pooled trials. Sensitivity analyses 1–2 were prespecified; 3–5 were post hoc.

We also examined conditional interaction estimates in the Bayesian regression model, focusing on effects for individuals with covariates set at reference values (age 60, BMI 25, no baseline comorbidities, baseline ordinal score 5, and sex predictors set between male and female values).

## Risk of bias assessment

Two investigators (T.L. and S.N.B.) assessed risk of bias associated with the effect of assignment to treatment on the primary outcome using Cochrane's Risk of Bias 2 tool [36], with disagreements resolved through discussion (S2 Table). This assessment was not used in the data synthesis.

## Registration

This study, including its statistical analysis plan (SAP), was registered with the International Prospective Register of Systematic Reviews (PROSPERO; registration number CRD42021254261) [37] prior to receiving patient data and amended prior to analyzing outcome data. The most significant amendment was broadening the primary outcome definition

from days 28–30 to 28–35 post enrollment to minimize missingness. Post-hoc changes to the analysis are shown in S3 Table. This study followed the Preferred Reporting Items for Systematic Reviews and Meta-analyses (PRISMA) extension for IPD analyses (PRISMA-IPD) [38].

## Results

### Study characteristics

Of 19 RCTs identified in our searches (18 from ClinicalTrials.gov; 1 from personal communication that was excluded due to lack of registration on ClinicalTrials.gov), 8 met final criteria for inclusion in our analysis (Fig 1): Outcomes Related to COVID-19 Treated With Hydroxychloroquine Among In-patients With Symptomatic Disease (ORCHID; NCT04332991) [14]; Treating COVID-19 With Hydroxychloroquine (TEACH; NCT04369742) [22]; Hydroxychloroquine vs. Azithromycin for Hospitalized Patients With Suspected or Confirmed COVID-19 (HAHPS; NCT04329832) [23, 39]; Washington University 352: Open-label, Randomized Controlled Trial of Hydroxychloroquine Alone or Hydroxychloroquine Plus Azithromycin or Chloroquine Alone or Chloroquine Plus Azithromycin in the Treatment of SARS CoV-2 Infection (WU352; NCT04341727); NCT04344444; A Randomized, Controlled Clinical Trial of the Safety and Efficacy of Hydroxychloroquine for the Treatment of COVID-19 in Hospitalized Patients (OAHU-COVID19; NCT04345692); NCT04335552; and Comparison Of Therapeutics for Hospitalized Patients Infected With SARS-CoV-2 In a Pragmatic aDaptive randoMizED Clinical Trial During the COVID-19 Pandemic (COVID MED; NCT04328012) (S4–S6 Tables) [40].

HCQ was a treatment arm in all studies; CQ was an additional treatment arm in one study (WU352). Comparators were placebo (3 trials), azithromycin (2 trials), and standard/usual care (2 trials); WU352 compared HCQ and CQ with and without azithromycin. HCQ dosing was usually (7 studies) 400 mg orally twice daily on day 1 and 200 mg twice daily on days 2–5, totaling 2,400 mg. Three trials were blinded; 5 were open-label.

Data on the prespecified primary outcome (the 7-point ordinal scale measured between days 28 and 30) was available for 90% of patients (695 out of 770). In the TEACH study, however, data was available for only 45% of patients (58 out of 128). Because of this, the decision was made to broaden the primary outcome window to days 28–35. This decision was made without examining the outcome data themselves. With the broader definition, primary outcome data was available for 76% of patients in TEACH (97 out of 128) and 95% of patients overall (734 out of 770).

### Risk of bias assessment

Risk of bias judgments are summarized for the primary outcome measurement in S2 Table. Overall, ORCHID and COVID MED were rated "low risk"; the other trials were rated "some concerns."

### Patient characteristics

Among 770 patients with laboratory-confirmed SARS-CoV-2 infection, 412 were randomized to HCQ/CQ treatment (398 HCQ; 14 CQ) and 358 to the control group (Table 1). Enrollment was at a median of 6 days (IQR, 3–8 days) after symptom onset. Most patients initiated dosing on the enrollment day.

Key baseline demographics were reasonably balanced between HCQ/CQ and control populations: mean age was 57 vs 55 years, male sex 59% vs 56%, White race 48% vs 44%, mean BMI 31.6 vs 33.2, mean comorbidities 3.16 vs 3.05 per patient, and mean ordinal score 4.1 vs 4.1,

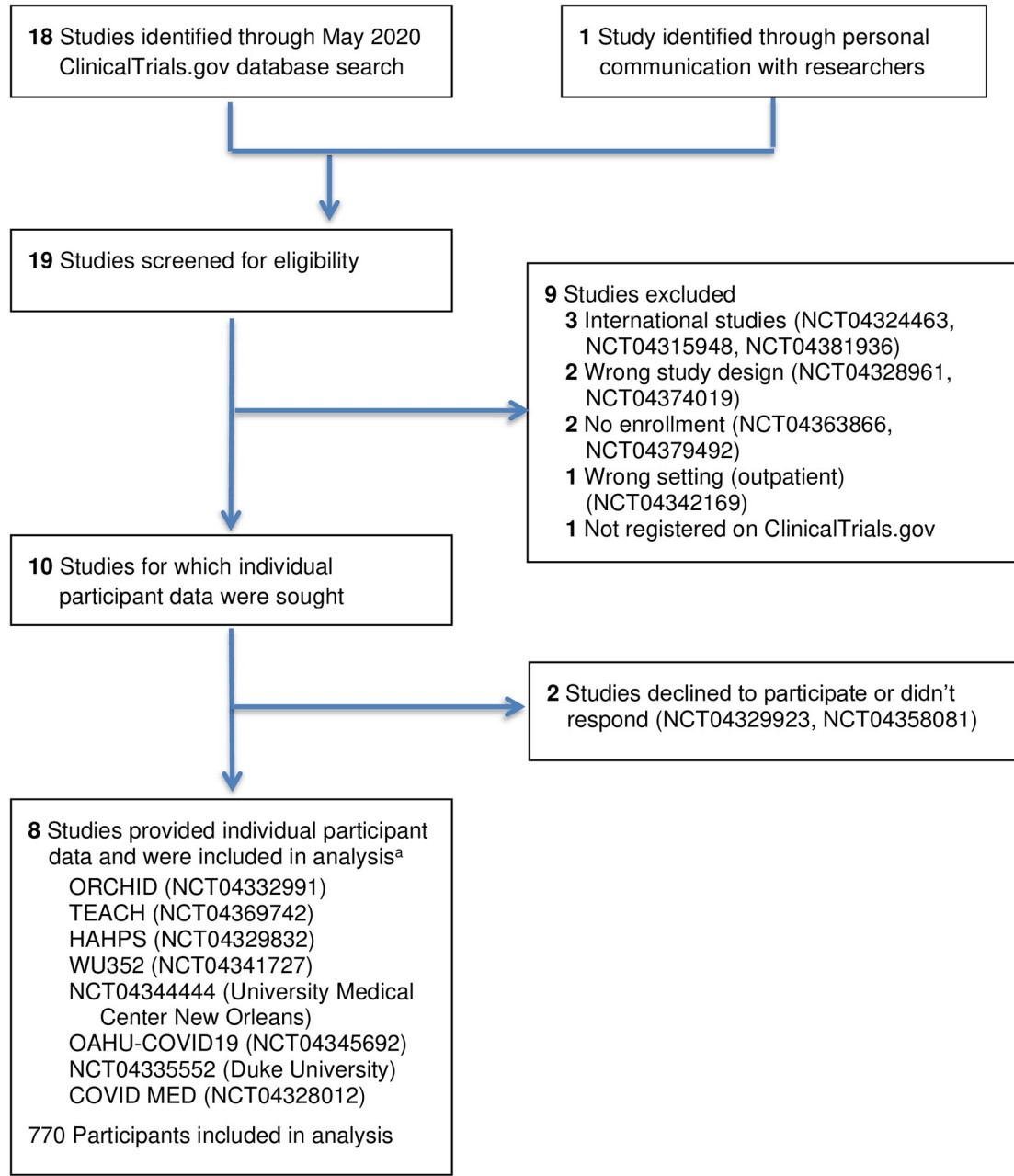

**Fig 1. Trial selection process.** [a]Two of the trials did not have study acronyms (only trial registration numbers). Abbreviations: COVID MED, Comparison Of Therapeutics for Hospitalized Patients Infected With SARS-CoV-2 In a Pragmatic aDaptive randoMizED Clinical Trial During the COVID-19 Pandemic; HAHPS, Hydroxychloroquine vs. Azithromycin for Hospitalized Patients With Suspected or Confirmed COVID-19; OAHU-COVID19, A Randomized, Controlled Clinical Trial of the Safety and Efficacy of Hydroxychloroquine for the Treatment of COVID-19 in Hospitalized Patients; ORCHID, Outcomes Related to COVID-19 Treated With Hydroxychloroquine Among In-patients With Symptomatic Disease; TEACH, Treating COVID-19 With Hydroxychloroquine; WU352, Washington University 352: Open-label, Randomized Controlled Trial of Hydroxychloroquine Alone or Hydroxychloroquine Plus Azithromycin or Chloroquine Alone or Chloroquine Plus Azithromycin in the Treatment of SARS CoV-2 Infection.

**Table 1. Participant characteristics overall and in each trial.**

| | Overall (n = 770) | | ORCHID (n = 479) | | TEACH (n = 128) | | HAHPS (n = 85) | | WU352 (n = 30) | NCT04344444 (n = 20) | | OAHU-COVID19 (n = 16) | | NCT04335552 (n = 11) | | COVID MED (n = 1) |
|---|---|---|---|---|---|---|---|---|---|---|---|---|---|---|---|---|
| | HCQ/CQ (n = 412) | Control (n = 358) | HCQ/CQ (n = 242) | Control (n = 237) | HCQ/CQ (n = 67) | Control (n = 61) | HCQ/CQ (n = 42) | Control (n = 43) | HCQ/CQ (n = 30) | HCQ/CQ (n = 15) | Control (n = 5) | HCQ/CQ (n = 10) | Control (n = 6) | HCQ/CQ (n = 6) | Control (n = 5) | Control (n = 1) |
| **Sex, No. (%)** | | | | | | | | | | | | | | | | |
| Female | 167 (41) | 158 (44) | 106 (44) | 105 (44) | 22 (33) | 30 (49) | 14 (33) | 19 (44) | 18 (60) | 2 (13) | 2 (40) | 4 (40) | 0 (0) | 1 (17) | 2 (40) | 0 (0) |
| Male | 244 (59) | 200 (56) | 135 (56) | 132 (56) | 45 (67) | 31 (51) | 28 (67) | 24 (56) | 12 (40) | 13 (87) | 3 (60) | 6 (60) | 6 (100) | 5 (83) | 3 (60) | 1 (100) |
| Missing/unknown | 1 (0) | 0 (0) | 1 (0) | 0 (0) | 0 (0) | 0 (0) | 0 (0) | 0 (0) | 0 (0) | 0 (0) | 0 (0) | 0 (0) | 0 (0) | 0 (0) | 0 (0) | 0 (0) |
| **Race, No. (%)** | | | | | | | | | | | | | | | | |
| Black | 95 (23) | 71 (20) | 58 (24) | 57 (24) | 14 (21) | 11 (18) | 0 (0) | 1 (2) | 18 (60) | 5 (33) | 2 (40) | 0 (0) | 0 (0) | 0 (0) | 0 (0) | 0 (0) |
| White | 197 (48) | 158 (44) | 109 (45) | 103 (43) | 32 (48) | 23 (38) | 29 (69) | 26 (60) | 11 (37) | 10 (67) | 2 (40) | 2 (20) | 0 (0) | 4 (67) | 3 (60) | 1 (100) |
| Multiple | 3 (1) | 3 (1) | 2 (1) | 3 (1) | 0 (0) | 0 (0) | 0 (0) | 0 (0) | 0 (0) | 0 (0) | 0 (0) | 0 (0) | 0 (0) | 1 (17) | 0 (0) | 0 (0) |
| Other[a] | 107 (26) | 113 (32) | 73 (30) | 74 (31) | 14 (21) | 18 (30) | 10 (24) | 13 (30) | 1 (3) | 0 (0) | 0 (0) | 8 (80) | 6 (100) | 1 (17) | 2 (40) | 0 (0) |
| Unavailable | 10 (2) | 13 (4) | 0 (0) | 0 (0) | 7 (10) | 9 (15) | 3 (7) | 3 (7) | 0 (0) | 0 (0) | 1 (20) | 0 (0) | 0 (0) | 0 (0) | 0 (0) | 0 (0) |
| **Ethnicity, No. (%)** | | | | | | | | | | | | | | | | |
| Hispanic | 144 (35) | 135 (38) | 91 (38) | 87 (37) | 25 (37) | 25 (41) | 15 (36) | 17 (40) | 1 (3) | 9 (60) | 3 (60) | 1 (10) | 0 (0) | 2 (33) | 3 (60) | 0 (0) |
| Not Hispanic | 248 (60) | 210 (59) | 145 (60) | 143 (60) | 42 (63) | 36 (59) | 27 (64) | 24 (56) | 25 (83) | 0 (0) | 0 (0) | 9 (90) | 6 (100) | 0 (0) | 0 (0) | 1 (100) |
| Unavailable | 20 (5) | 13 (4) | 6 (2) | 7 (3) | 0 (0) | 0 (0) | 0 (0) | 2 (5) | 4 (13) | 6 (40) | 2 (40) | 0 (0) | 0 (0) | 4 (67) | 2 (40) | 0 (0) |
| Age (5y bins), median (IQR) | 55.0 (45.0–70.0) | 55.0 (45.0–65.0) | 55.0 (45.0–65.0) | 55.0 (40.0–65.0) | 65.0 (55.0–75.0) | 65.0 (55.0–75.0) | 55.0 (40.0–65.0) | 50.0 (40.0–60.0) | 55.0 (45.0–60.0) | 70.0 (62.5–75.0) | 65.0 (60.0–65.0) | 67.5 (56.3–70.0) | 45.0 (41.3–56.3) | 47.5 (41.3–61.3) | 55.0 (50.0–60.0) | 55.0 (55.0–55.0) |
| **BMI** | | | | | | | | | | | | | | | | |
| Median (IQR) | 30.0 (25.7–36.1) | 31.4 (27.0–37.2) | 31.3 (26.4–37.2) | 31.1 (27.2–36.5) | 25.9 (22.9–30.5) | 29.3 (24.9–35.9) | 31.7 (26.6–37.4) | 36.3 (30.7–41.0) | 30.5 (28.0–34.0) | 27.7 (24.3–34.1) | 29.3 (25.4–34.2) | 28.8 (26.7–34.2) | 26.0 (24.0–30.5) | 33.6 (30.2–36.4) | 44.2 (34.5–47.9) | 37.9 (37.9–37.9) |
| Missing, No. (%) | 16 (4) | 19 (5) | 16 (7) | 18 (8) | 0 (0) | 0 (0) | 0 (0) | 1 (2) | 0 (0) | 0 (0) | 0 (0) | 0 (0) | 0 (0) | 0 (0) | 0 (0) | 0 (0) |
| **Baseline ordinal scale, No. (%)** | | | | | | | | | | | | | | | | |
| 2: hosp, mech vent | 23 (6) | 25 (7) | 13 (5) | 19 (8) | 0 (0) | 0 (0) | 7 (17) | 6 (14) | 0 (0) | 0 (0) | 0 (0) | 3 (30) | 0 (0) | 0 (0) | 0 (0) | 0 (0) |
| 3: hosp, NIV | 49 (12) | 42 (12) | 28 (12) | 27 (11) | 13 (19) | 7 (11) | 6 (14) | 7 (16) | 1 (3) | 1 (7) | 1 (20) | 0 (0) | 0 (0) | 0 (0) | 0 (0) | 0 (0) |
| 4: hosp, supp ox | 191 (46) | 175 (49) | 116 (48) | 108 (46) | 25 (37) | 34 (56) | 23 (55) | 24 (56) | 13 (43) | 7 (47) | 3 (60) | 6 (60) | 5 (83) | 1 (17) | 0 (0) | 1 (100) |
| 5: hosp, no ox | 146 (35) | 112 (31) | 85 (35) | 83 (35) | 26 (39) | 17 (28) | 6 (14) | 6 (14) | 16 (53) | 7 (47) | 0 (0) | 1 (10) | 1 (17) | 5 (83) | 5 (100) | 0 (0) |
| Missing | 3 (1) | 4 (1) | 0 (0) | 0 (0) | 3 (4) | 3 (5) | 0 (0) | 0 (0) | 0 (0) | 0 (0) | 1 (20) | 0 (0) | 0 (0) | 0 (0) | 0 (0) | 0 (0) |
| **Baseline ordinal scale (numeric)** | | | | | | | | | | | | | | | | |
| Mean (SD) | 4.1 (0.8) | 4.1 (0.8) | 4.1 (0.8) | 4.1 (0.9) | 4.2 (0.8) | 4.2 (0.6) | 3.7 (0.9) | 3.7 (0.9) | 4.5 (0.6) | 4.4 (0.6) | 3.8 (0.5) | 3.5 (1.1) | 4.2 (0.4) | 4.8 (0.4) | 5.0 (0.0) | 4.0 (NA) |

*(Continued)*

**Table 1.** (Continued)

| | Overall (n = 770) | | ORCHID (n = 479) | | TEACH (n = 128) | | HAHPS (n = 85) | | WU352 (n = 30) | NCT0434444 (n = 20) | | OAHU-COVID19 (n = 16) | | NCT0435552 (n = 11) | | COVID MED (n = 1) |
|---|---|---|---|---|---|---|---|---|---|---|---|---|---|---|---|---|
| | HCQ/CQ (n = 412) | Control (n = 358) | HCQ/CQ (n = 242) | Control (n = 237) | HCQ/CQ (n = 67) | Control (n = 61) | HCQ/CQ (n = 42) | Control (n = 43) | HCQ/CQ (n = 30) | HCQ/CQ (n = 15) | Control (n = 5) | HCQ/CQ (n = 10) | Control (n = 6) | HCQ/CQ (n = 6) | Control (n = 5) | Control (n = 1) |
| Missing, No. (%) | 3 (1) | 4 (1) | 0 (0) | 0 (0) | 3 (4) | 3 (5) | 0 (0) | 0 (0) | 0 (0) | 0 (0) | 1 (20) | 0 (0) | 0 (0) | 0 (0) | 0 (0) | 0 (0) |
| **Days between symptom onset and enrollment** | | | | | | | | | | | | | | | | |
| Median (IQR) | 5.0 (3.0–8.0) | 6.0 (3.0–8.0) | 5.0 (3.0–7.0) | 5.0 (3.0–7.0) | 7.0 (3.0–9.0) | 7.0 (4.0–14.0) | 8.0 (5.3–12.0) | 9.0 (7.0–11.0) | 5.0 (3.3–9.8) | 2.0 (2.0–5.5) | 6.0 (2.0–10.0) | 4.5 (4.0–6.5) | 2.5 (0.5–3.8) | NA | NA | 7.0 (7.0–7.0) |
| Missing, No. (%) | 6 (1) | 5 (1) | 0 (0) | 0 (0) | 0 (0) | 0 (0) | 0 (0) | 0 (0) | 0 (0) | 0 (0) | 0 (0) | 0 (0) | 0 (0) | 6 (100) | 5 (100) | 0 (0) |
| **Baseline comorbidity count** | | | | | | | | | | | | | | | | |
| Median (IQR) | 3.0 (2.0–4.0) | 3.0 (2.0–4.0) | 3.0 (2.0–4.0) | 3.0 (2.0–4.0) | 3.0 (3.0–4.0) | 4.0 (3.0–5.0) | 2.0 (1.0–2.0) | 2.0 (1.0–3.0) | 3.0 (2.0–4.0) | 5.0 (4.0–5.5) | 3.0 (3.0–4.0) | 2.0 (2.0–3.0) | 2.0 (1.3–4.3) | NA | NA | NA |
| Missing, No. (%) | 16 (4) | 15 (4) | 3 (1) | 2 (1) | 3 (4) | 2 (3) | 4 (10) | 5 (12) | 0 (0) | 0 (0) | 0 (0) | 0 (0) | 0 (0) | 6 (100) | 5 (100) | 1 (100) |
| **Azithromycin use (at or before d28), No. (%)** | | | | | | | | | | | | | | | | |
| Not assigned, did not take | 311 (75) | 248 (69) | 195 (81) | 193 (81) | 54 (81) | 44 (72) | 28 (67) | 0 (0) | 16 (53) | 4 (27) | 5 (100) | 10 (100) | 4 (67) | 4 (67) | 2 (40) | 0 (0) |
| Not assigned, took | 75 (18) | 63 (18) | 47 (19) | 44 (19) | 13 (19) | 17 (28) | 14 (33) | 0 (0) | 0 (0) | 1 (7) | 0 (0) | 0 (0) | 2 (33) | 0 (0) | 0 (0) | 0 (0) |
| Assigned, took | 26 (6) | 45 (13) | 0 (0) | 0 (0) | 0 (0) | 0 (0) | 0 (0) | 42 (98) | 14 (47) | 10 (67) | 0 (0) | 0 (0) | 0 (0) | 2 (33) | 3 (60) | 0 (0) |
| Assigned, did not take | 0 (0) | 1 (0) | 0 (0) | 0 (0) | 0 (0) | 0 (0) | 0 (0) | 1 (2) | 0 (0) | 0 (0) | 0 (0) | 0 (0) | 0 (0) | 0 (0) | 0 (0) | 0 (0) |
| Missing | 0 (0) | 1 (0) | 0 (0) | 0 (0) | 0 (0) | 0 (0) | 0 (0) | 0 (0) | 0 (0) | 0 (0) | 0 (0) | 0 (0) | 0 (0) | 0 (0) | 0 (0) | 1 (100) |
| **Concurrent corticosteroid use (at or before d28), No. (%)** | | | | | | | | | | | | | | | | |
| Yes | 57 (14) | 61 (17) | 39 (16) | 49 (21) | 7 (10) | 6 (10) | 7 (17) | 6 (14) | 2 (7) | 2 (13) | 0 (0) | 0 (0) | 0 (0) | 0 (0) | 0 (0) | 0 (0) |
| Missing | 6 (1) | 5 (1) | 0 (0) | 0 (0) | 0 (0) | 0 (0) | 0 (0) | 0 (0) | 0 (0) | 0 (0) | 0 (0) | 0 (0) | 0 (0) | 6 (100) | 5 (100) | 0 (0) |
| **First dose received on day of enrollment, No. (%)** | | | | | | | | | | | | | | | | |
| Yes | 386 (94) | 324 (91) | 241 (100) | 225 (95) | 50 (75) | 51 (84) | 41 (98) | 31 (72) | 29 (97) | 13 (87) | 5 (100) | 6 (60) | 6 (100) | 6 (100) | 5 (100) | 1 (100) |

(*Continued*)

**Table 1.** (Continued)

| | Overall (n = 770) | | ORCHID (n = 479) | | TEACH (n = 128) | | HAHPS (n = 85) | | WU352 (n = 30) | NCT0434444 (n = 20) | | OAHU-COVID19 (n = 16) | | NCT04335552 (n = 11) | | COVID MED (n = 1) |
|---|---|---|---|---|---|---|---|---|---|---|---|---|---|---|---|---|
| | HCQ/CQ (n = 412) | Control (n = 358) | HCQ/CQ (n = 242) | Control (n = 237) | HCQ/CQ (n = 67) | Control (n = 61) | HCQ/CQ (n = 42) | Control (n = 43) | HCQ/CQ (n = 30) | HCQ/CQ (n = 15) | Control (n = 5) | HCQ/CQ (n = 10) | Control (n = 6) | HCQ/CQ (n = 6) | Control (n = 5) | Control (n = 1) |
| Missing | 6 (1) | 12 (3) | 0 (0) | 9 (4) | 4 (6) | 2 (3) | 1 (2) | 1 (2) | 1 (3) | 0 (0) | 0 (0) | 0 (0) | 0 (0) | 0 (0) | 0 (0) | 0 (0) |

Abbreviations: BMI, body mass index; HCQ/CQ, hydroxychloroquine or chloroquine; IQR, interquartile range; NIV, noninvasive ventilation (includes BiPAP/CPAP and/or high-flow oxygen).

[a]Includes American Indian or Alaska Native, Asian, Native Hawaiian or Other Pacific Islander, and other. To protect participant privacy, ORCHID's data set grouped three of its race variables with low frequencies (American Indian or Alaska Native, Asian, Native Hawaiian or Other Pacific Islander). For the sake of uniformity, we combined these groups and the "Other" category for the other studies as well.

respectively. Post-baseline use of corticosteroids was 14% vs 17% and of azithromycin 24.5% vs 30.2%, respectively. Six patients with BMI values <10 or >70 were deemed probable recording errors and treated as missing in primary and mortality analyses.

## Primary outcome: Pooled and subgroup analysis

The standardized proportional odds ratio (OR) for ordinal score at 28–35 days was 0.97 (95% credible interval [CrI], 0.76–1.24); the corresponding unadjusted proportional OR was 0.98 (95% CI, 0.75–1.28) (Fig 2 and Table 2). These results are consistent with no effect of HCQ/ CQ. We found no appreciable heterogeneity in estimated treatment-study interactions among the 8 studies after adjusting for individual-level baseline covariates ($\tau$ = 0.87 on the log odds scale; 95% CrI, 0.01–5.17); tests for publication bias were inconclusive (S1 Fig). There were no substantial effects of HCQ/CQ observed in any prespecified subgroup, nor in a post-hoc subgroup analysis based on the time between symptom onset and enrollment (Fig 2; S5 Fig). We investigated potential trends across strata of baseline ordinal score and BMI (Fig 3). We examined corresponding conditional effect estimates and found insufficient evidence to conclude that an effect of HCQ/CQ on the primary outcome differs by BMI or baseline ordinal score, after adjusting for other baseline covariates. These and other conditional effect analyses are shown in S2 Fig.

## Mortality: Pooled and subgroup analysis

Mortality at 28–35 days was similar in HCQ/CQ vs control groups (10%, n = 43 HCQ/CQ vs 9%, n = 34 control; model-adjusted risk difference [RD], -0.01 [95% CrI, -0.04 to 0.02]; plug-in RD, -0.01 [95% CI, -0.06 to 0.04], where a positive RD favors HCQ/CQ) (Fig 4). Again, we observed no appreciable heterogeneity in treatment effect estimates across prespecified subgroups. On the RD scale, there was greater uncertainty about the effect of HCQ/CQ upon mortality for those with higher baseline risk scores. For those with low baseline risk scores, the model precisely predicts only small effects of HCQ/CQ (RD for the first group, -0.01 [95% CrI, -0.02 to 0.01]; second group, -0.01 [95% CrI, -0.02 to 0.01]; third group, 0.00 [95% CrI, -0.02 to

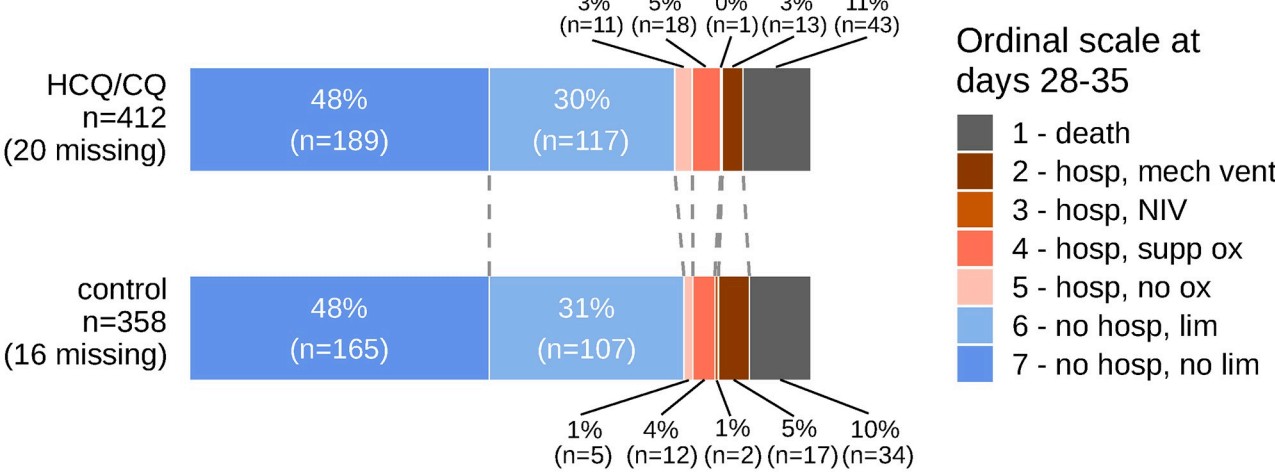

**Fig 2. Primary outcome data by treatment group.** Scores were defined as: (1) death; (2) hospitalized, on mechanical ventilation or ECMO; (3) hospitalized, on non-invasive ventilation (BiPAP/CPAP and/or high-flow oxygen); (4) hospitalized, requiring oxygen; (5) hospitalized, not requiring oxygen; (6) not hospitalized, with limitation; and (7) not hospitalized, without limitations. Abbreviations: HCQ/CQ, hydroxychloroquine or chloroquine; NIV, noninvasive ventilation.

**Table 2. Primary, secondary, and safety outcomes, overall and by trial.**

**Primary outcome: Ordinal scale improvement under HCQ/CQ at day 28–35[a]**

| | Overall (n = 770) | ORCHID (n = 479) | TEACH (n = 128) | HAHPS (n = 85) | WU352 (n = 30) | NCT04344444 (n = 20) | OAHU-COVID19 (n = 16) | NCT0435552 (n = 11) | COVID MED (n = 1) | Missing |
|---|---|---|---|---|---|---|---|---|---|---|
| Model-standardized proportional OR (95% CrI) | 0.97 (0.76 to 1.24) | 0.96 (0.74 to 1.23) | 1.00 (0.68 to 1.59) | 0.92 (0.61 to 1.36) | 1.01 (0.51 to 1.72) | 0.83 (0.52 to 1.59) | 0.76 (0.38 to 1.30) | 1.27 (0.66 to 2.44) | 1.26 (0.23 to 4.03) | NA |
| Plug-in proportional OR (95% CI) | 0.98 (0.75 to 1.28) | 1.02 (0.73 to 1.43) | 0.75 (0.35 to 1.60) | 0.81 (0.36 to 1.81) | NA | 0.31 (0.01 to 2.85) | 0.33 (0.04 to 2.30) | 1.38 (0.11 to 18.47) | NA | 36 (4.7%) |

**Secondary outcomes[a]**

| | Overall (n = 770) | ORCHID (n = 479) | TEACH (n = 128) | HAHPS (n = 85) | WU352 (n = 30) | NCT04344444 (n = 20) | OAHU-COVID19 (n = 16) | NCT0435552 (n = 11) | COVID MED (n = 1) | Missing |
|---|---|---|---|---|---|---|---|---|---|---|
| Mortality at day 28–35 under HCQ/CQ vs control, model-standardized RD (95% CrI) | -0.01 (-0.04 to 0.02) | -0.01 (-0.04 to 0.02) | 0.00 (-0.04 to 0.05) | -0.03 (-0.07 to 0.05) | 0.00 (-0.02 to 0.03) | -0.02 (-0.07 to 0.04) | -0.04 (-0.12 to 0.02) | 0.01 (-0.03 to 0.08) | 0.01 (-0.16 to 0.09) | NA |
| Mortality at day 28–35 under HCQ/CQ vs control, plug-in RD (95% CI) | -0.01 (-0.06 to 0.04) | 0.00 (-0.05 to 0.06) | -0.01 (-0.16 to 0.14) | -0.12 (-0.26 to 0.02) | NA | 0.13 (-0.37 to 0.64) | -0.20 (-0.58 to 0.18) | -0.08 (-0.73 to 0.57) | NA | 36 (4.7%) |

| | Overall Control (n = 358) | Overall HCQ/CQ (n = 412) | ORCHID Control (n = 237) | ORCHID HCQ/CQ (n = 242) | TEACH Control (n = 61) | TEACH HCQ/CQ (n = 67) | HAHPS Control (n = 43) | HAHPS HCQ/CQ (n = 42) | WU352 HCQ/CQ (n = 30) | NCT04344444 Control (n = 5) | NCT04344444 HCQ/CQ (n = 15) | OAHU-COVID19 Control (n = 6) | OAHU-COVID19 HCQ/CQ (n = 10) | NCT0435552 Control (n = 5) | NCT0435552 HCQ/CQ (n = 6) | COVID MED Control (n = 1) | Missing |
|---|---|---|---|---|---|---|---|---|---|---|---|---|---|---|---|---|---|
| Days of hospitalization between enrollment and day 28 (median) | 7 | 7 | 8 | 7 | 5 | 5 | 6 | 6.5 | 4 | 2 | 9 | 9 | 18 | 8.5 | 4.5 | 9 | 1 |
| Patients on mechanical ventilation between enrollment and day 28, No. (%) | 76 (21) | 82 (20) | 58 (24) | 51 (21) | 4 (7) | 7 (10) | 11 (28) | 14 (35) | 3 (10) | 0 (0) | 1 (7) | 2 (33) | 5 (50) | 1 (25) | 1 (17) | 0 (0) | 7 |
| **Safety outcomes** | | | | | | | | | | | | | | | | | |
| Adverse events (AEs), count (per patient) | 104 (0.29) | 160 (0.39) | 39 (0.16) | 50 (0.21) | 59 (0.97) | 63 (0.94) | 2 (0.05) | 3 (0.07) | 29 (0.97) | 1 (0.20) | 8 (0.53) | 3 (0.50) | 7 (0.70) | NA | NA | 9 | 11 |
| Serious adverse events (SAEs), count (per patient) | 32 (0.09) | 53 (0.13) | 12 (0.05) | 18 (0.07) | 11 (0.18) | 14 (0.21) | 0 | 0 | 9 (0.30) | 1 (0.25) | 4 (0.27) | 2 (0.33) | 4 (0.40) | 6 (1.20) | 4 (0.67) | 0 | 1 |
| QTc prolongation AEs, count (per patient) | 8 (0.02) | 14 (0.03) | 3 (0.01) | 2 (0.01) | 1 (0.02) | 3 (0.04) | 2 (0.05) | 3 (0.07) | 2 (0.07) | 1 (0.20) | 3 (0.20) | 1 (0.17) | 1 (0.10) | NA | NA | 0 (0) | 11 |
| QTc prolongation SAEs, count (per patient) | 1 (0.00) | 1 (0.00) | 0 | 0 | 0 | 1 (0.01) | 0 | 0 | 0 | 0 | 0 | 1 (0.17) | 0 | NA | NA | 0 | 11 |
| Elevated LFTs AEs, count (per patient) | 4 (0.01) | 21 (0.05) | 3 (0.01) | 12 (0.05) | 0 | 1 (0.01) | 0 | 0 | 0 | 0 | 7 (0.47) | 1 (0.17) | 1 (0.10) | NA | NA | 0 | 11 |
| Elevated LFTs SAEs, count (per patient) | 0 | 0 | 0 | 0 | 0 | 0 | 0 | 0 | 0 | 0 | 0 | 0 | 0 | NA | NA | 0 | 11 |
| Arrhythmia AEs, count (per patient) | 10 (0.03) | 8 (0.02) | 10 (0.04) | 1 (0.00) | 0 | 0 | 0 | 0 | 2 (0.07) | 0 | 4 (0.27) | 0 | 1 (0.10) | NA | NA | 0 | 11 |
| Arrhythmia SAEs, count (per patient) | 3 (0.01) | 1 (0.00) | 3 (0.01) | 0 | 0 | 0 | 0 | 0 | 0 | 0 | 1 (0.07) | 0 | 0 | NA | NA | 0 | 11 |

**Missingness in the primary outcome (ordinal scale between day 28–35)**

*(Continued)*

**Table 2.** (Continued)

| Nonmissing primary outcome, No. (%) | 342 (96) | 392 (95) | 237 (100) | 242 (100) | 47 (77) | 50 (75) | 42 (98) | 42 (100) | 27 (90) | 5 (100) | 15 (100) | 6 (100) | 10 (100) | 4 (80) | 6 (100) | 1 (100) | NA |

[a]Positive odds ratios and risk differences favor HCQ/CQ over control.

Abbreviations: 95% CrI, 95% credible intervals; HCQ/CQ, hydroxychloroquine or chloroquine; LFTs, liver function tests; NA, not applicable; OR, odds ratio; RD, risk difference.

### Ordinal scale at days 28-35

| Subgroup | Proportional OR, model-standardized (95% CrI) | Proportional OR, plug-in (95% CI) |
|---|---|---|
| Overall (n=770) | 0.97 (0.76 to 1.24) | 0.98 (0.75 to 1.28) |
| **Study** | | |
| ORCHID (n=479) | 0.96 (0.74 to 1.23) | 1.02 (0.73 to 1.43) |
| TEACH (n=128) | 1.00 (0.68 to 1.59) | 0.75 (0.35 to 1.60) |
| HAHPS (n=85) | 0.92 (0.61 to 1.36) | 0.81 (0.36 to 1.81) |
| WU352 (n=30) | 1.01 (0.51 to 1.72) | NA |
| NCT04344444 (n=20) | 0.83 (0.52 to 1.59) | 0.31 (0.01 to 2.85) |
| OAHU-COVID19 (n=16) | 0.76 (0.38 to 1.30) | 0.33 (0.04 to 2.30) |
| NCT04335552 (n=11) | 1.27 (0.66 to 2.44) | 1.38 (0.11 to 18.47) |
| COVID MED (n=1) | 1.26 (0.23 to 4.03) | NA |
| **Baseline ordinal scale** | | |
| 2: hosp, mech vent (n=48) | 0.63 (0.28 to 1.57) | 0.58 (0.20 to 1.61) |
| 3: hosp, NIV (n=91) | 0.75 (0.41 to 1.44) | 0.79 (0.37 to 1.67) |
| 4: hosp, supp ox (n=366) | 0.94 (0.66 to 1.50) | 0.91 (0.61 to 1.36) |
| 5: hosp, no ox (n=258) | 1.13 (0.63 to 1.71) | 0.97 (0.57 to 1.63) |
| **Age** | | |
| ≤29 (n=40) | 0.53 (0.22 to 1.17) | 0.80 (0.23 to 2.87) |
| 30-49 (n=198) | 1.10 (0.76 to 1.62) | 0.81 (0.46 to 1.44) |
| 50-69 (n=330) | 0.90 (0.67 to 1.19) | 1.38 (0.91 to 2.09) |
| 70-79 (n=124) | 0.76 (0.53 to 1.19) | 0.49 (0.24 to 0.97) |
| 80+ (n=78) | 1.62 (0.84 to 3.07) | 1.82 (0.79 to 4.26) |
| **Sex** | | |
| Female (n=325) | 0.97 (0.73 to 1.40) | 1.09 (0.71 to 1.65) |
| Male (n=444) | 0.91 (0.69 to 1.33) | 0.92 (0.65 to 1.32) |
| **No. of comorbidities** | | |
| 0 (n=4) | 0.77 (0.08 to 5.29) | NA |
| 1 (n=57) | 0.81 (0.37 to 1.64) | 1.11 (0.41 to 3.03) |
| 2 (n=208) | 1.01 (0.70 to 1.58) | 1.31 (0.76 to 2.24) |
| 3 (n=213) | 1.09 (0.77 to 1.51) | 0.84 (0.49 to 1.41) |
| ≥4 (n=257) | 0.83 (0.55 to 1.16) | 0.92 (0.58 to 1.46) |
| **Body mass index** | | |
| ≤20 (n=22) | 1.40 (0.56 to 4.70) | 0.30 (0.03 to 1.89) |
| 20-25 (n=118) | 0.74 (0.48 to 1.16) | 0.97 (0.47 to 1.99) |
| 25-30 (n=200) | 0.72 (0.55 to 1.04) | 0.67 (0.39 to 1.15) |
| 30-35 (n=168) | 0.87 (0.64 to 1.17) | 1.19 (0.68 to 2.10) |
| ≥35 (n=221) | 1.47 (0.93 to 2.16) | 1.33 (0.80 to 2.22) |
| **Baseline risk group** | | |
| 1st (n=154) | 0.80 (0.43 to 1.48) | 0.62 (0.30 to 1.26) |
| 2nd (n=154) | 0.91 (0.61 to 1.27) | 1.11 (0.56 to 2.18) |
| 3rd (n=154) | 1.01 (0.67 to 1.45) | 0.97 (0.52 to 1.81) |
| 4th (n=154) | 1.01 (0.71 to 1.48) | 0.89 (0.49 to 1.59) |
| 5th (n=154) | 0.94 (0.59 to 1.49) | 1.08 (0.61 to 1.92) |

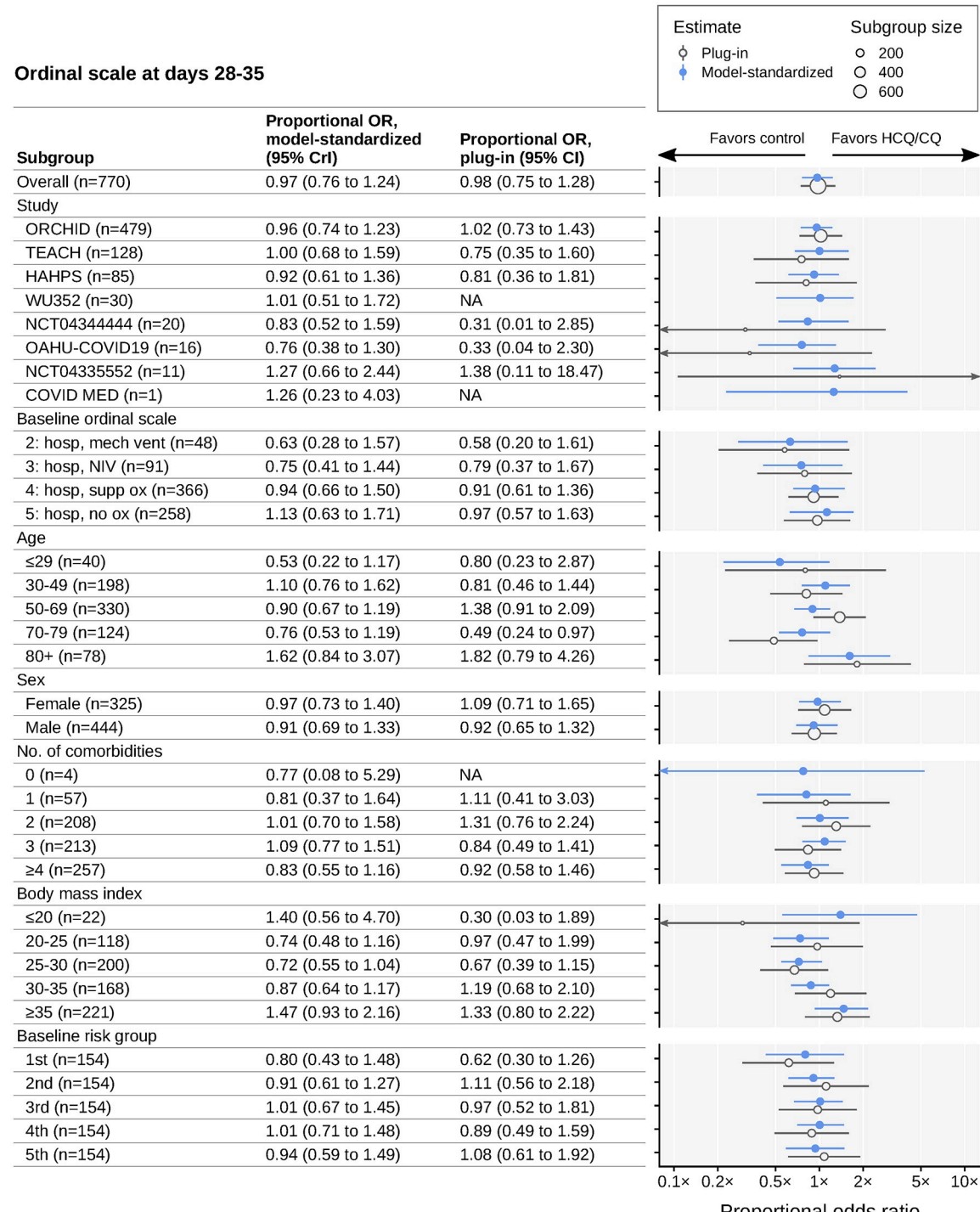

**Fig 3. Subgroup analysis of differences in ordinal scale at days 28–35.** Estimated proportional odds ratios comparing day 28–35 ordinal scale in HCQ/CQ versus control groups. Estimates are given for the pooled patient population and for subgroups. Blue circles represent model-standardized estimates; blue horizontal lines represent 95% credible intervals. Open grey circles represent plug-in estimates; grey horizontal lines represent 95% confidence intervals. Grey circle size represents the number of patients in the corresponding subgroup. Arrows indicate uncertainty intervals extending beyond plot limits. Study acronyms are explained in Fig 1. Two trials did not have study acronyms (only trial registration numbers). Abbreviations: 95% CrI, 95% credible intervals; HCQ/CQ, hydroxychloroquine or chloroquine; NA, not applicable; NIV, noninvasive ventilation (includes BiPAP/CPAP and/or high-flow oxygen); OR, odds ratio.

**Mortality at days 28-35**

| Subgroup | Risk difference, model-standardized (95% CrI) | Risk difference, plug-in (95% CI) |
|---|---|---|
| Overall (n=770) | -0.01 (-0.04 to 0.02) | -0.01 (-0.06 to 0.04) |
| Study | | |
| ORCHID (n=479) | -0.01 (-0.04 to 0.02) | 0.00 (-0.05 to 0.06) |
| TEACH (n=128) | 0.00 (-0.04 to 0.05) | -0.01 (-0.16 to 0.14) |
| HAHPS (n=85) | -0.03 (-0.07 to 0.05) | -0.12 (-0.26 to 0.02) |
| WU352 (n=30) | 0.00 (-0.02 to 0.03) | NA |
| NCT04344444 (n=20) | -0.02 (-0.07 to 0.04) | 0.13 (-0.37 to 0.64) |
| OAHU-COVID19 (n=16) | -0.04 (-0.12 to 0.02) | -0.20 (-0.58 to 0.18) |
| NCT04335552 (n=11) | 0.01 (-0.03 to 0.08) | -0.08 (-0.73 to 0.57) |
| COVID MED (n=1) | 0.01 (-0.16 to 0.09) | NA |
| Baseline ordinal scale | | |
| 2: hosp, mech vent (n=48) | -0.11 (-0.29 to 0.08) | -0.19 (-0.49 to 0.10) |
| 3: hosp, NIV (n=91) | -0.06 (-0.19 to 0.08) | -0.11 (-0.32 to 0.09) |
| 4: hosp, supp ox (n=366) | 0.00 (-0.03 to 0.03) | 0.00 (-0.06 to 0.05) |
| 5: hosp, no ox (n=258) | 0.01 (-0.01 to 0.04) | 0.04 (-0.03 to 0.11) |
| Age | | |
| ≤29 (n=40) | -0.03 (-0.09 to 0.01) | -0.06 (-0.22 to 0.11) |
| 30-49 (n=198) | 0.00 (-0.02 to 0.03) | 0.00 (0.00 to 0.00) |
| 50-69 (n=330) | -0.02 (-0.05 to 0.01) | 0.01 (-0.07 to 0.08) |
| 70-79 (n=124) | -0.05 (-0.10 to 0.02) | -0.13 (-0.27 to 0.02) |
| 80+ (n=78) | 0.06 (-0.03 to 0.15) | 0.17 (-0.07 to 0.41) |
| Sex | | |
| Female (n=325) | -0.01 (-0.04 to 0.03) | 0.01 (-0.06 to 0.08) |
| Male (n=444) | -0.01 (-0.05 to 0.03) | -0.02 (-0.08 to 0.04) |
| No. of comorbidities | | |
| 0 (n=4) | -0.03 (-0.26 to 0.27) | 0.00 (0.00 to 0.00) |
| 1 (n=57) | -0.04 (-0.09 to 0.04) | -0.08 (-0.22 to 0.06) |
| 2 (n=208) | -0.01 (-0.04 to 0.03) | -0.03 (-0.10 to 0.04) |
| 3 (n=213) | 0.00 (-0.03 to 0.04) | 0.04 (-0.05 to 0.12) |
| ≥4 (n=257) | -0.03 (-0.08 to 0.03) | 0.00 (-0.10 to 0.10) |
| Body mass index | | |
| ≤20 (n=22) | 0.04 (-0.04 to 0.17) | -0.15 (-0.47 to 0.16) |
| 20-25 (n=118) | -0.02 (-0.06 to 0.02) | 0.01 (-0.12 to 0.14) |
| 25-30 (n=200) | -0.04 (-0.09 to 0.00) | -0.05 (-0.16 to 0.05) |
| 30-35 (n=168) | -0.03 (-0.06 to 0.01) | 0.03 (-0.07 to 0.14) |
| ≥35 (n=221) | 0.03 (-0.01 to 0.06) | 0.01 (-0.07 to 0.10) |
| Baseline risk group | | |
| 1st (n=154) | -0.01 (-0.02 to 0.01) | 0.01 (-0.03 to 0.06) |
| 2nd (n=154) | -0.01 (-0.02 to 0.01) | 0.02 (-0.05 to 0.08) |
| 3rd (n=154) | 0.00 (-0.02 to 0.02) | 0.00 (-0.06 to 0.07) |
| 4th (n=154) | -0.01 (-0.05 to 0.03) | -0.05 (-0.17 to 0.07) |
| 5th (n=154) | -0.03 (-0.15 to 0.07) | -0.03 (-0.19 to 0.13) |

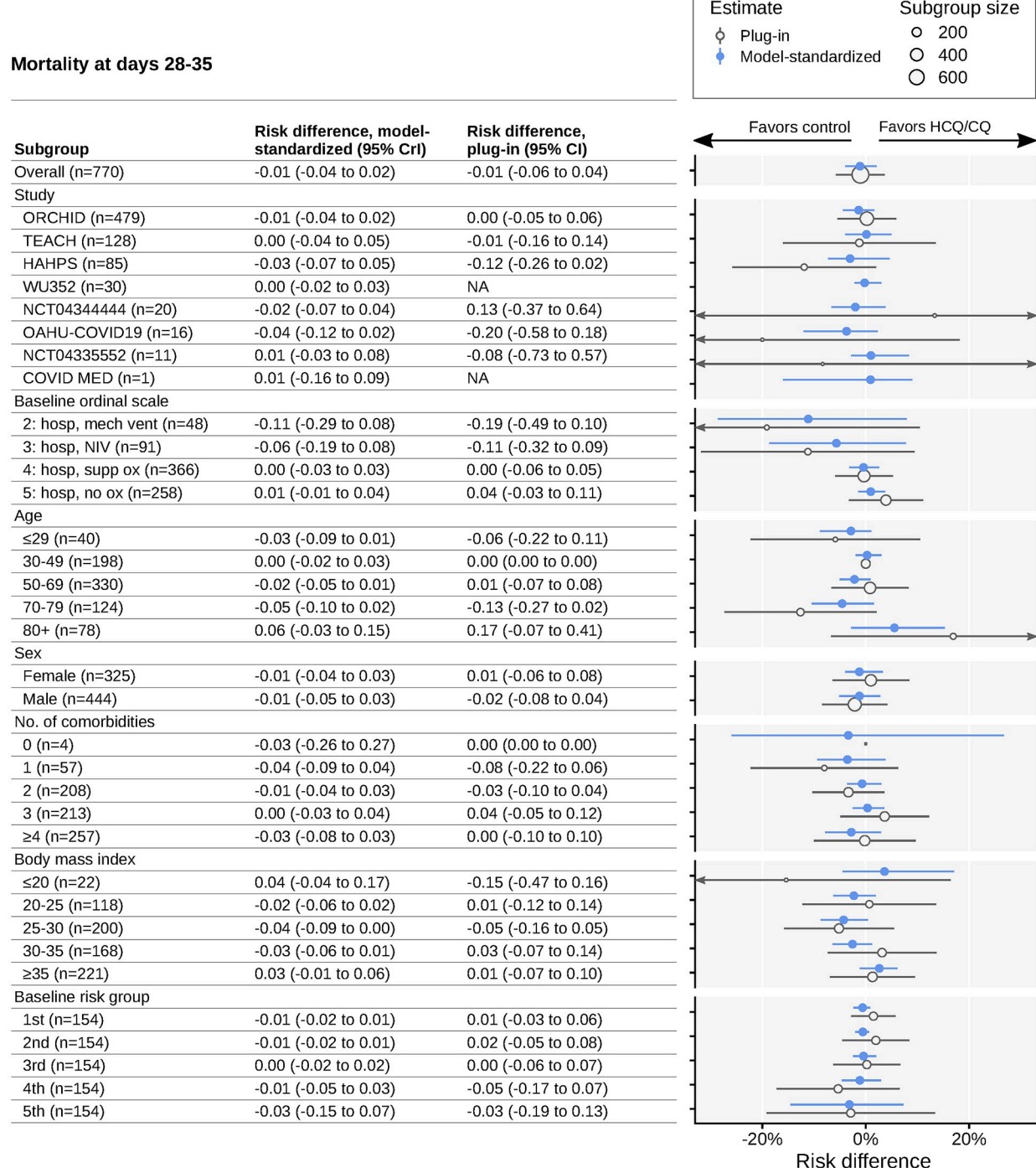

**Fig 4. Subgroup analysis of differences in mortality at days 28–35.** Estimated risk differences for day 28–35 mortality in HCQ/CQ versus control groups. Estimates are given for the pooled patient population and for subgroups. Blue circles represent model-standardized estimates; blue horizontal lines represent 95% credible intervals. Open grey circles represent plug-in estimates; grey horizontal lines represent 95% CIs. Grey circle size represents the number of patients in the corresponding subgroup. Arrows indicate uncertainty intervals extending beyond plot limits. Study acronyms are explained in Fig 1. Two trials did not have study acronyms (only trial registration numbers). Abbreviations: 95% CrI, 95% credible intervals; HCQ/CQ, hydroxychloroquine or chloroquine; NA, not applicable; NIV, noninvasive ventilation (includes BiPAP/CPAP and/ or high-flow oxygen); OR, odds ratio.

0.02]; fourth group, -0.01 [95% CrI, -0.05 to 0.03]; and fifth group, -0.03 [95% CrI, -0.15 to 0.07]) (Fig 4). Separate estimates of mortality at day 28–35 under control and HCQ/CQ are shown in S3 Fig, with conditional effect analyses in S2 Fig.

## Sensitivity analyses

Our alternative models and outcome definitions produced qualitatively similar conclusions about overall and subgroup effects. Simpler models than the one we prespecified (removing interactions, adding weakly informative priors) had better leave-one-out cross-validation performance than our primary prespecified model, while the model fit only to ORCHID data performed worse than our primary model (S2 Appendix). Our primary model fit without individual-level treatment-covariate interactions yielded an adjusted OR of 0.92 (95% CrI, 0.58–1.41) for the benefit of HCQ/CQ. Posterior predictive checks of our primary model indicated good in-sample fit (S4 Fig).

## Exploratory analysis of secondary and safety outcomes

There were similar rates of mechanical ventilation between enrollment and day 28 (20% [n = 82] HCQ/CQ vs 21% [n = 76] control). HCQ/CQ and control patients had a median post-enrollment hospital length of stay of 7 days.

Overall AE rates were numerically higher in the HCQ/CQ vs the control group (0.39 vs 0.29 per patient, respectively), as were overall SAE rates (0.13 vs 0.09 per patient). These comparisons are potentially vulnerable to aggregation bias, but are broadly consistent with results from within each of the largest three studies (ORCHID, TEACH, and HAHPS). LFTs elevation AE rates were also numerically higher with HCQ/CQ (0.05 [n = 21] vs 0.01 [n = 4] per patient). QTc prolongation and arrhythmia AE and SAE rates were similar (Table 2).

## Discussion

This IPD meta-analysis of 8 RCTs in 770 hospitalized COVID-19 patients comparing HCQ/CQ and control treatment confirms results of at least 16 published RCTs showing no benefit of HCQ/CQ [14, 16–30]. Neither the primary outcome measurement, ordinal scale at 28–35 days, nor the secondary outcome measurement, mortality at 28–35 days, was improved with HCQ/CQ in the pooled study population. We found no subgroup in which appreciable benefits could be observed for the primary outcome. Overall rates of AEs and SAEs, and elevated LFT AEs, but not QTc prolongation AEs, were higher with HCQ/CQ than controls.

This study adds value to the literature by synthesizing IPD from 8 RCTs, of which 7 were terminated early, 4 unpublished, and 4 published individually and/or in aggregate data meta-analyses (ORCHID, TEACH, HAHPS, and NCT04335552) [14, 22, 23, 39, 40]. To our knowledge, this is the first published meta-analysis of HCQ/CQ trials in hospitalized COVID-19 patients to use IPD rather than aggregate data. We are aware of only 4 additional published IPD meta-analyses for COVID-19 therapeutics (3 planned; 1 smaller one completed) [41–44]. Two meta-analyses planned to use IPD data to supplement aggregate data but could not obtain them [45, 46].

This analysis used a random effects model to synthesize IPD about the efficacy of HCQ/CQ overall and stratified by 6 key covariates. Pooling individual-level data allowed us to obtain relatively stable estimates of patient-level subgroup effects [47] that have only been minimally investigated in other published aggregate data meta-analyses [40, 46, 48].

We produced more precise estimates of the overall effects relative to a similar analysis based on ORCHID alone, which represented 62% of our study sample; subgroup effect estimates also tended to be more precise. Our model was flexibly specified, and our qualitative

conclusions were robust to both alternative model specifications and varying outcome time windows.

It is instructive to consider our results in the context of these drugs' history in COVID-19.

Inconsistent antiviral activity was reported for HCQ/CQ in vitro and in animal models against coronaviruses including SARS-CoV-2 [2, 49–51]. Early in the pandemic, faster SARS-CoV-2 shedding clearance was reported in small, mainly uncontrolled trials, prompting recommendations by some experts and guidelines to administer HCQ/CQ empirically in COVID-19 [1, 3, 52]. Based partly on these data, and safety in other indications, the FDA issued an HCQ/CQ EUA for hospitalized COVID-19 patients on March 28, 2020. The accelerated viral shedding clearance reported by Gautret et al. [3], however, could not be replicated [53].

Increased rates of QTc prolongation were reported from small trials, particularly with higher dosing and azithromycin co-administration, though torsades de pointes was rare [5–7, 54]. These reports and others prompted FDA issuance of a Drug Safety Communication on April 24, 2020, about HCQ/CQ "known risks" [55].

Subsequent retrospective-observational studies in hospitalized COVID-19 patients did not find evidence of benefit, some reporting potential harm [8–10, 15]. Additional mostly smaller retrospective-observational studies reported decreased mortality with HCQ monotherapy and with azithromycin [11–13, 56]. These studies were observational, without randomization or blinding, and with baseline imbalances and other potential sources of confounding.

In May 2020, the FDA published a pharmacovigilance memorandum on HCQ/CQ safety data [57]. QT prolongation and increased LFTs were the most common cardiac and non-cardiac SAEs, respectively. Our study similarly observed increased rates of LFTs elevation AEs with HCQ/CQ but not QTc prolongation. Case reports of HCQ-induced liver injury were published before and during the pandemic [58–60]. Exclusion of patients with prolonged QTc in all of our trials and of co-administration of additional QTc-prolonging medications in some may have averted QTc prolongation in our population.

Results from at least five RCTs became available in the spring/summer of 2020.

The first was a Chinese multicenter open-label RCT in 150 mild-moderate severity hospitalized COVID-19 patients [16]. Mean interval from symptom onset was 16.6 days (longer than our 6.2 days). The main outcome, 28-day nasopharyngeal swab conversion, was similar in HCQ (85.4%) versus standard care patients (81.3%). AEs occurred in 30% vs 9%; SAEs in 2 vs 0 patients, respectively. No arrhythmias or QTc prolongation occurred. The pattern of safety results in our study was similar.

Results from the open-label RECOVERY platform trial comparing HCQ and standard care in hospitalized COVID-19 patients were released publicly (with HCQ arm enrollment cessation) on June 5, 2020, and published soon after [17]. Median duration from symptom onset was 9 days (longer than our 6 days). RECOVERY patients were sicker than those in our study (17% vs 6% received mechanical ventilation and/or ECMO). The trial's primary outcome, 28-day mortality, occurred in 421 of 1,561 (27%) HCQ vs 790 of 3,155 (25%) standard care patients (rate ratio, 1.09; 95% CI, 0.97–1.23). No evidence of benefit was reported in any subgroup. Adverse safety signals included higher cardiac death rates but not arrhythmias. Our efficacy overall and subgroup results were similar. Results from RECOVERY (and other emerging data) led the FDA on June 15, 2020, to revoke its EUA for HCQ/CQ.

ORCHID, one of the studies in this IPD meta-analysis, was a double-blind, placebo-controlled trial of HCQ in 479 hospitalized COVID-19 patients. On June 20, 2020, the NIH announced discontinuation of enrollment after the data and safety monitoring board "determined that while there was no harm, the study drug was very unlikely to be beneficial" [61]. Results were published on November 9, 2020 [14]. Median interval from symptom onset was 5

days (similar to our pooled cohort). The primary outcome, the WHO 14-day ordinal score, was similar between HCQ vs placebo (adjusted proportional OR, 1.02; 95% CI, 0.73–1.42), as was the 28-day score (adjusted proportional OR, 1.07; 95% CI, 0.54–2.09). No evidence of benefit was reported for any subgroup. Overall AE (21% vs 16%), overall SAE (7% vs 5%), and QTc prolongation (5.9% vs 3.3%) rates were numerically higher with HCQ. Our safety results are similar, except for QTc prolongation. Our efficacy analysis focuses on a different estimand but produced qualitatively similar conclusions. Our primary analysis produced a comparably precise estimate of the overall effect of HCQ on day 28–35 ordinal scale compared with a similar analysis using a model fit to ORCHID data alone (OR 1.00; CrI 0.79 to 1.26). Subgroup effects from the full-data model tended to be more precise than those from the ORCHID-only model.

A July 4, 2020, WHO press release stated that its open-labeled SOLIDARITY trial (fourth RCT) would stop enrollment in its HCQ arm after an interim analysis showed little or no reduction in mortality; safety signals were also detected [62]. Results were published on December 2, 2020 [19]. Baseline ordinal score distribution was similar to our study. The primary outcome, in-hospital mortality, occurred in 104 of 947 (11.0%) HCQ vs 84 of 906 (9.3%) control patients (rate ratio, 1.19; 95% CI, 0.89–1.59).

Coalition Covid-19 Brazil I was a multicenter, open-label trial that compared HCQ, HCQ and azithromycin, and standard care in 504 mild-moderate severity hospitalized COVID-19 patients [18]. HCQ was started at a median of 7 days after symptom onset (similar to our study). The primary outcome, a 15-day ordinal scale, was not significantly different for HCQ or HCQ/azithromycin versus standard care. QTc prolongation and LFTs elevation were more common with HCQ (the latter in line with our study).

Updated NIH and Infectious Diseases Society of America COVID-19 guidelines (June 2020) recommended against HCQ/CQ use except in clinical trials [63, 64].

Eleven additional RCTs assessing HCQ/CQ in hospitalized patients with COVID-19 were published later in the pandemic (winter 2020 and 2021). All showed no substantial treatment benefit or worse primary outcomes, including clinical ordinal scales, mortality, composite scores, and viral shedding; some showed adverse safety signals—increased QTc prolongation, acute kidney injury, and AEs/SAEs [20–30].

At least 50 aggregate data meta-analyses evaluating HCQ/CQ in hospitalized COVID-19 patients have been published, with the overwhelming majority finding lack of evidence of convincing clinical benefit, and many finding worse clinical outcomes and increased AE and SAE rates [40, 46]. In contrast with our IPD meta-analysis, these studies included either no or minimal patient-level subgroup analyses.

For other COVID-19 populations (outpatients, prophylaxis), most studies did not find evidence of HCQ benefit [65–76].

The results of our study are congruous with the thrust of HCQ/CQ studies thus far, which have showed equivocal preclinical efficacy and no convincing evidence of clinical efficacy, as well as adverse safety signals, in the overwhelming majority of retrospective-observational studies, RCTs, and meta-analyses [5–30].

Our study has some key limitations. First, we included trials with open-label designs and varying treatments (HCQ vs CQ; with and without azithromycin). Second, 6 studies had some risk of bias. Third, we pooled a limited set of studies identified in our searches because some principal investigators declined participation and we excluded international trials. Fourth, we made SAP modifications after PROSPERO registration (S3 Table). Fifth, our analysis combined HCQ and CQ arms; only 16 patients received CQ alone. Sixth, in our safety analysis we did not harmonize AE and SAE definitions among the included studies. Seventh, our final study search was conducted in June 2020. This timeline is consistent with what is typical for

IPD meta-analyses, which commonly take upwards of two years to complete [31]. Repeating our searches of ClinicalTrials.gov in July 2022 and applying the same eligibility criteria revealed two unpublished studies (NCT04429867 and NCT04458948) potentially eligible for inclusion registered in June and July of 2020, respectively. (Registrations of new trials of hydroxychloroquine for COVID-19 had slowed substantially by August 2020 [77].

## Conclusions

Our IPD meta-analysis confirms published results from retrospective-observational studies, RCTs, and aggregate data meta-analyses showing no strong evidence of efficacy, but concerning safety signals, for hydroxychloroquine (or chloroquine) use overall and in prespecified subgroups of hospitalized COVID-19 patients.

## Supporting information

**S1 Appendix. PRISMA-IPD checklist of items to include when reporting a systematic review and meta-analysis of individual participant data (IPD).**
(PDF)

**S2 Appendix. Methods supplement: Description of the primary outcome model and estimands.**
(PDF)

**S3 Appendix. Search strategy in detail.**
(PDF)

**S1 Table. Data dictionary from data harmonization spreadsheet.**
(PDF)

**S2 Table. Risk of bias assessment.**
(PDF)

**S3 Table. Changes to the prespecified statistical analysis plan.**
(PDF)

**S4 Table. Primary, secondary, and safety outcomes.**
(PDF)

**S5 Table. Trial characteristics: Treatment groups, participant assessment, and inclusion/ exclusion criteria.**
(PDF)

**S6 Table. Merging trial arms.**
(PDF)

**S1 Fig. Assessing between-study heterogeneity.** Estimated study coefficients from the primary outcome model, with 66% and 95% credible intervals (CrI) indicated. "Control" terms represent differences in predicted outcomes between sites under control, for individuals with similar baseline covariates, on the proportional log-odds scale. Treatment effect terms represent differences in predicted treatment effect between sites, again for individuals with similar baseline covariates. Positive coefficients represent, respectively, better ordinal scale outcomes under control and benefit of HCQ/CQ at days 28–35 post-enrollment. Estimated between-study standard deviations were 1.48 for the "Control" terms (95% CrI, 0.16–5.98; on the log-odds scale) and 0.87 for the treatment effects (95% CrI, 0.01–5.17). Tests of publication bias applied to unadjusted study-specific proportional ORs from the 6 studies with patients

assigned to both HCQ/CQ and control yielded inconclusive results (Egger's test: p = 0.10; Begg's test: p = 0.57); these should be interpreted with caution given the small number of included studies. HCQ/CQ indicates hydroxychloroquine or chloroquine.
(PDF)

**S2 Fig. Conditional covariate effects.** Conditional effects from the Bayesian proportional odds model. Shown are (1) the relative risk of mechanical ventilation or death at day 28–35; (2) the estimated probabilities of mechanical ventilation or death at day 28–35 under control and HCQ/CQ; and (3) the log proportional odds ratio comparing HCQ/CQ and control. Each of these effects are shown for reference individuals with the following covariate values: age 60, BMI 25, no baseline comorbidities, baseline ordinal score of 5, and sex coefficient set between male and female values. Curves for continuous covariates are accompanied by 50% and 95% credible bands; intervals for discrete covariates are accompanied by 66% and 95% credible intervals. BMI indicates body mass index; HCQ/CQ, hydroxychloroquine or chloroquine.
(PDF)

**S3 Fig. Estimated mortality rate in subgroups under both control and HCQ/CQ.** Shown are both plug-in estimates (based on the proportion of deaths in each subgroup) along with 95% CIs, and model-adjusted estimates with 95% credible intervals. The model used is the same as for the primary outcome analysis. HCQ/CQ indicates hydroxychloroquine or chloroquine.
(PDF)

**S4 Fig. Posterior predictive check of primary outcomes by study.** Shown are observed outcome data (first row) and draws for the posterior predictive distribution (subsequent rows) of the ordinal outcome scale at day 28–35, plotted against the expected linear predictor for each individual. Each column corresponds to one study in our analysis. Data points have been jittered for clarity.
(PDF)

**S5 Fig. Exploratory analysis of time between symptom onset and enrollment.** Post-hoc exploratory analysis of subgroups based on time between symptom onset and enrollment. Subgroups are based on approximate tertiles. Shown are (A) proportional odds ratios from models fit by maximum likelihood within each subgroup, with 95% CIs; (B) the empirical risk of survival for each treatment group within each subgroup, with 95% CIs; and (C) empirical risk differences within each subgroup, with 95% CIs. HCQ/CQ indicates hydroxychloroquine or chloroquine.
(PDF)

## Acknowledgments

The names of our nonauthor collaborators, including investigators and support staff for the trials analyzed, are listed below. We thank Emily Bartlett of Johns Hopkins University for manuscript preparation assistance. We also are grateful for the many contributions made by members of the Trial Innovation Network and the COVID-19 Collaboration Platform. This collaboration is based on research using data from TEACH, HAHPS, WU352, NCT04344444, OAHU-COVID19, NCT04335552, and COVID MED that has been made available through Vivli, Inc. Vivli has not contributed to or approved, and is not in any way responsible for, the contents of this publication.

The following individuals within the Pandemic Response COVID-19 Research Collaboration Platform for HCQ/CQ Pooled Analyses were instrumental in the planning and conduct of this study at each of the participating institutions:

**ORCHID (NCT04332991):** Derek C. Angus, MD, MPH; Alexandra Weissman, MD, MPH; Donald M. Yealy, MD (University of Pittsburgh, Pittsburgh, Pennsylvania). Roy G. Brower, MD (Johns Hopkins University School of Medicine, Baltimore, Maryland). Samuel M. Brown, MD, MS; Lindsay M. Leither, DO (Intermountain Medical Center and University of Utah, Salt Lake City, Utah). Jonathan D. Casey, MD, MSc; Sean P. Collins, MD; Todd W. Rice MD (Vanderbilt University Medical Center, Nashville, Tennessee). Steven Y. Chang, MD, PhD (UCLA, Los Angeles, California). John C. Eppensteiner, MD (Duke University, Durham, North Carolina). Michael R. Filbin, MD; Douglas L. Hayden, PhD; David A. Schoenfeld, PhD; B. Taylor Thompson, MD; Christine A. Ulysse, MS (Massachusetts General Hospital, Boston, Massachusetts). D. Clark Files, MD; Kevin W. Gibbs, MD (Wake Forest School of Medicine, Winston-Salem, North Carolina). Adit A. Ginde, MD, MPH; Marc Moss, MD (University of Colorado School of Medicine, Aurora, Colorado). Michelle N. Gong, MD, MS (Albert Einstein College of Medicine, Montefiore Medical Center, Bronx, New York). Frank E. Harrell, Jr, PhD; Christopher J. Lindsell, PhD (Vanderbilt University School of Medicine, Nashville, Tennessee). Catherine L. Hough, MD, MSc; Akram Khan, MD (Oregon Health and Science University School of Medicine, Portland, Oregon). Nicholas J. Johnson, MD; Bryce R. H. Robinson, MD, MS (University of Washington, Seattle, Washington). Michael A. Matthay, MD (University of California, San Francisco, California). Pauline K. Park, MD (University of Michigan, Ann Arbor, Michigan). Nathan I. Shapiro MD, MPH (Beth Israel Deaconess Medical Center and Harvard Medical School, Boston, Massachusetts). Jay S. Steingrub, MD (University of Massachusetts Medical School-Baystate, Springfield, Massachusetts).

**TEACH (NCT04369742):** Jonathan S. Austrian, MD, NYU; Ellie Carmody, MD, MPH; Camila Delgado PhD; Yanina Dubrovskaya, PharmD, BCIDP; Jaishvi Eapen, MD; Brooklyn Henderson, RN; Alexander Hrycko, MD; Morris Jrada, MD; Yi Li, PhD; Prithiv J. Prasad, MBBS; Vanessa Raabe, MD; Gabriel A. Robbins, MD; Andrea B. Troxel, PhD (NYU Grossman School of Medicine, New York, New York). Martin Bäcker, MD; Dinuli Delpachitra, MBBS (NYU Long Island School of Medicine, Mineola, New York). Jack A. DeHovitz, MD, MPH, MHCDS, FACP (State University of New York Downstate Health Sciences University, Brooklyn, New York).

**HAHPS (NCT04329832):** Jacki Anderson; Brent Armbruster, BS; Valerie Aston, MBA; Katie Brown, BSN; Allison M Butler, MS; Briana Crook, BS; Diana Grant, BSN; Danielle Groat, PhD; Prathyusha Kodakandla; Naresh Kumar, MPH; Lindsay Leither, DO; Heather Maestas, BS; Mardee Merrill, BS; Amanda Nelson; Rilee Smith, MPH; Rajendu Srivastava, MD, MPH; Nathan Starr, DO; Brandon J Webb, MD (Intermountain Healthcare, Murray, Utah). Macy Barrios, BS; Tom Greene, PhD; Jorden Greer, BS; Colin K Grissom, MD; Benjamin Haaland, PhD; Estelle Harris, MD; Stacy Johnson, MD; Robert Paine III, MD; Ithan Peltan, MD, MSc; Lisa Weaver, BS; Jian Ying, PhD (University of Utah, Salt Lake City, Utah).

**WU352 (NCT04341727):** Mansi Agarwal; Megan Arb; Teresa Arb; Andrea Patterson Brown; Jennifer Bruns; Kelly Caplice; PeChaz Clark; Rachel Cody; Brittany Depp; Charles Goss; Zackary Jakuboski; Michael Klebert; Kathleen McNulty; Tina Nolte; Arnita Pitts; Rachel Presti; Shea Roesel-Wakeland; Tara Sattler; Andrej Spec; Heather Wilkins; Brittney Zwijack (Washington University School of Medicine, St Louis, Missouri). Kristopher Bakos; Kathryn Vehe (Barnes-Jewish Hospital, St Louis, Missouri).

**NCT04344444:** Andrew Chapple, PhD; Michael Hagensee, MD, PhD; Amber Trauth, MPH; Brianne Voros, MS (Louisiana State University Health Sciences Center, New Orleans, Louisiana). Jyotsna Fuloria, MD (University Medical Center-New Orleans, New Orleans, Louisiana).

**OAHU-COVID19 (NCT04345692):** Todd B. Seto, MD (The Queen's Medical Center, Honolulu, Hawaii).

**NCT04335552:** Arthur W. Baker, MD, MPH; Thuy Le, MD, PhD; Ahmad Mourad, MD; Susanna Naggie, MD, MHS; Shanti Narayanasami, MBBS; Sean M. O'Brien, PhD; Nwora L. Okeke, MD, MPH; Frank W. Rockhold, PhD; Robert J. Rolfe Jr., MD; Nicholas A. Turner, MD, MHS; Rebekah Wrenn, PharmD (Duke University Medical Center, Durham, North Carolina).

**COVID MED (NCT04328012):** Anne M. Gadomski, MD, MPH, FAAP (Bassett Research Institute and Bassett Medical Center, Cooperstown, NY).

## Author Contributions

**Conceptualization:** Leon Di Stefano, Elizabeth L. Ogburn, Daniel O. Scharfstein, Tianjing Li, Barbara E. Bierer, Daniel F. Hanley, Daniel Freilich.

**Data curation:** Leon Di Stefano, Malathi Ram.

**Formal analysis:** Leon Di Stefano, Elizabeth L. Ogburn, Daniel O. Scharfstein, Tianjing Li, Daniel Freilich.

**Funding acquisition:** Daniel F. Hanley.

**Investigation:** Yussef Bennani, Samuel M. Brown, Whitney R. Buckel, Meredith E. Clement, Mark J. Mulligan, Jane A. O'Halloran, Adriana M. Rauseo, Wesley H. Self, Matthew W. Semler, Todd Seto, Jason E. Stout, Robert J. Ulrich, Jennifer Victory, Daniel Freilich.

**Methodology:** Leon Di Stefano, Daniel O. Scharfstein, Daniel Freilich.

**Project administration:** Malathi Ram, Preeti Khanal, Nichol McBee, Marianne R. Gildea, Daniel Freilich.

**Resources:** Leon Di Stefano, Elizabeth L. Ogburn, Daniel O. Scharfstein, Tianjing Li, Barbara E. Bierer, Daniel F. Hanley, Daniel Freilich.

**Software:** Leon Di Stefano.

**Supervision:** Elizabeth L. Ogburn, Daniel O. Scharfstein, Barbara E. Bierer, Daniel F. Hanley, Daniel Freilich.

**Validation:** Leon Di Stefano, Daniel O. Scharfstein, Daniel Freilich.

**Visualization:** Leon Di Stefano, Megan R. Clark.

**Writing – original draft:** Leon Di Stefano, Malathi Ram, Daniel O. Scharfstein, Tianjing Li, Preeti Khanal, Sheriza N. Baksh, Daniel Freilich.

**Writing – review & editing:** Leon Di Stefano, Elizabeth L. Ogburn, Malathi Ram, Daniel O. Scharfstein, Tianjing Li, Preeti Khanal, Sheriza N. Baksh, Nichol McBee, Joshua Gruber, Marianne R. Gildea, Megan R. Clark, Neil A. Goldenberg, Yussef Bennani, Samuel M. Brown, Whitney R. Buckel, Meredith E. Clement, Mark J. Mulligan, Jane A. O'Halloran, Adriana M. Rauseo, Wesley H. Self, Matthew W. Semler, Todd Seto, Jason E. Stout, Robert J. Ulrich, Jennifer Victory, Barbara E. Bierer, Daniel F. Hanley, Daniel Freilich.

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
