## [Decision Letter · Decision Letter 0]

27 May 2022

PONE-D-22-06839Hydroxychloroquine/Chloroquine for the Treatment of Hospitalized Patients with COVID-19: An Individual Participant Data Meta-AnalysisPLOS ONE

Dear Dr. Freilich,

Thank you for submitting your manuscript to PLOS ONE. After careful consideration, we feel that it has merit but does not fully meet PLOS ONE’s publication criteria as it currently stands. Therefore, we invite you to submit a revised version of the manuscript that addresses the points raised during the review process.

ACADEMIC EDITOR: Would you please go through the comments raised by the diligent reviewers and amend the review accordingly. 

We look forward to receiving your revised manuscript.

Kind regards,

A. M. Abd El-Aty

Academic Editor

PLOS ONE

Journal Requirements:

The authors have read the journal’s policy and have the following competing interests: SNB, NM, MRC, and DFH reported receiving research funding from the Department of Defense for clinical trials of convalescent plasma for COVID-19 outside the submitted work. NAG reported receiving salary support from the National Institutes of Health (NIH) National Center for Advancing Translational Sciences via a Johns Hopkins Clinical and Translational Science Award outside the submitted work. YB reported being a site investigator for Janssen outside the submitted work. SMB reported service as chair of a data and safety monitoring board for a Hamilton clinical trial in respiratory failure; fees paid to Intermountain Healthcare from Faron Pharmaceuticals and Sedana Pharmaceuticals for steering committee service for a clinical trial in acute respiratory distress syndrome; research grants to Intermountain Healthcare from Janssen, NIH, Centers for Disease Control and Prevention, and Department of Defense; and royalties from Oxford University Press and Brigham Young University, outside the submitted work. MEC reported service on a Roche advisory board and as a site investigator for Janssen outside the submitted work. This does not alter the authors’ adherence to PLOS ONE policies on sharing data and materials. All other authors report no conflicts of interest.

3. We noted in your submission details that a portion of your manuscript may have been presented or published elsewhere. Please clarify whether this publication was peer-reviewed and formally published. If this work was previously peer-reviewed and published, in the cover letter please provide the reason that this work does not constitute dual publication and should be included in the current manuscript.

4. One of the noted authors is a group or consortium [insert name of group or team]. In addition to naming the author group, please list the individual authors and affiliations within this group in the acknowledgments section of your manuscript. Please also indicate clearly a lead author for this group along with a contact email address.’ 

5. We note that you have referenced (ie. Bewick et al. [5]) which has currently not yet been accepted for publication. Please remove this from your References and amend this to state in the body of your manuscript: (ie “Bewick et al. [Unpublished]”) as detailed online in our guide for authors

Reviewers' comments:

Reviewer's Responses to Questions

**Comments to the Author**

1. Is the manuscript technically sound, and do the data support the conclusions?

Reviewer #1: Yes

Reviewer #2: Yes

Reviewer #3: Partly

Reviewer #4: Yes

2. Has the statistical analysis been performed appropriately and rigorously? 

Reviewer #1: Yes

Reviewer #2: Yes

Reviewer #3: No

Reviewer #4: Yes

3. Have the authors made all data underlying the findings in their manuscript fully available?

Reviewer #1: Yes

Reviewer #2: Yes

Reviewer #3: Yes

Reviewer #4: Yes

4. Is the manuscript presented in an intelligible fashion and written in standard English?

Reviewer #1: Yes

Reviewer #2: Yes

Reviewer #3: Yes

Reviewer #4: Yes

5. Review Comments to the Author

Reviewer #1: Thanks for inviting me to reviewing this paper. This is a high-quality IPD meta-analysis. I especially enjoy the statistical methods used for the analysis. Although the topic has already been reported by several other meta-analyses, it still brings value through the individual participant data perspective— the evidence could be ranked at the top over previous similar meta-analyses.

I only have two questions, about the safety assessment. I see the author lists and found there some renewed methodologist in safety assessment. They may have their own consideration. But I think a further investigation may makes the paper more attractive.

1. First, the authors seem failed to define the AEs and SAEs clearly. Ambiguous definition raises confusions during data extraction and impacts the transparency as well as reproducibility.

2. Second, the authors reported the adverse events outcomes descriptively, this is not the best solution because without weighting scheme the results tend to at risk of Simpson’s paradox. To avoid this, I suggest the authors statistically analysis the safety outcomes, for example, the GLMM/Beta-Binominal model.

Reviewer #2: I read the manuscript carefully. It’s a well-written and robust manuscript only there are some minor comments.

Minor comments:

1-Please add reference for PRISMA guide line.

2- the authors claimed that their study stick to PRISMA guide line. However, based on this protocol, search strategy should be presented in text or supplementary file. Also, PRISMA check list is necessary.

Reviewer #3: Dear Editor,

I carefully read the manuscript by Di Stefano et al.

My comments and suggestions for the authors are the following:

- The search process should be updated (this is a critical issue!).

- The authors mention that their meta-analysis is PRISMA compliant. However, they should specify the version of PRISMA guidelines they referred to and include the article among the references of the manuscript.

- Line 212: Including in a meta-analysis data from a personal communication is not a recomendable and increase the risk of publication biases for the analysis.

- The authors should perform also Begg and Egger's tests.

- The authors should consider to refer to doi: 10.1093/ehjcvp/pvaa105 in the discussion of the manuscript.

Reviewer #4: The authors present a very interesting IPD meta-analysis based on 8 RCTs including unanalyzed data from early trials on the efficacy and safety of HCQ/CQ in COVID-19 prevention and treatment. The paper and analysis are overall of very good quality. I suggested some minors restructuration and additional subgroup analysis that could help strengthen the authors’ claim.

Abstract

Authors did not mention what was the primary outcome in the method (they just mentioned the metric of the scale). It is important to know upfront the main measure of their work (at least the conceptual framing behind it). The authors mentioned again the “COVID-19 ordinal scores” in results without giving a specific measure in the abstract. They could conclude on the potential harming effects of HCQ/CQ.

Introduction

I really appreciate the transparency protocol the authors put into place (though it would be great to have the exact link for Vivli and CCP repositories to be able to consult the database). Outcomes are better explained in the main text; however, authors could justify why they used these specific levels. Was this recommendation base on the studies they re-analyzed? Or is this just a general classification recommended by health authorities and authors had to recode the whole thing? Precisions may help in that part. Also, authors mentioned a change in the outcome window due to missing data. How did the authors handle missing data in their study?

PLOS One is a very generalist journal so it would be great if the authors could recall the strength-weakness of RCTs vs. observational studies for a general audience as well as the benefits of randomization and how it helps to make causal inferences (i.e., controlling for confounding, random sampling criteria, etc. they could cite Schulz et al., 2002 or Vandenbroucke, 2004 for instance). This could also help strengthen the theoretical part of the paper which is unbalanced compared to the other sections of the paper.

Results

The authors mentioned early on that, due to missing data, they had to broaden days of post enrollment from 28-30 to 28-35. It would still be interesting though to check whether the main results changed in function of this amendment and how the authors can justify days 35 as the cut-off and not 34 or 33.

The authors make a compelling case of justifying many sub-group analyses to explore whether the HCQ/CQ presented local benefits for age, gender, BMI, etc. subgroups. Another possible and important subgroup is the timing of the medication. Some authors (see Prodomos & Rumschlag, 2020 or Million et al., 2020) have argued that HCQ is only effective when provided earlier, not associated with worsening disease and safe. Although I’m aware this argument is very borderline, if the authors have enough studies to compared early vs. late treatment, they would be able to respond to those critics and relativize some open-access analysis that claim early treatment showed better improvement (https://c19hcq.com/).

Discussion

The discussion part is much more detailed than the introduction. The authors provide a review of RCTs history on HCQ and COVID-19. I would recommend moving some parts of the discussion to the introduction to better contextualize the debate on HCQ and help naive readers understand what’s at stake and why their work is important.

The Annex and online supplementary materials are very complete and detailed.

6. PLOS authors have the option to publish the peer review history of their article (what does this mean?). If published, this will include your full peer review and any attached files.

Reviewer #1: **Yes: **Chang Xu

Reviewer #2: **Yes: **Bahman Razi

Reviewer #3: No

Reviewer #4: **Yes: **Jordane Boudesseul

---

## [Author Response · Author response to Decision Letter 0]

11 Jul 2022

Response to Academic Editor and Reviewers

Thank you for giving us the opportunity to revise our manuscript, “Hydroxychloroquine/chloroquine for the treatment of hospitalized patients with COVID-19: An individual participant data meta-analysis.” 

We sincerely appreciate the thorough and knowledgeable review. We have responded to the editor and reviewer concerns here and have tracked our edits in the revised files. We hope that these responses are satisfactory and that the revised manuscript will be acceptable for publication in PLOS ONE.

Journal Requirements

Response: We have reviewed the templates to ensure our manuscript meets PLOS ONE’s style requirements, and in particular have checked our file names.

The authors have read the journal’s policy and have the following competing interests: SNB, NM, MRC, and DFH reported receiving research funding from the Department of Defense for clinical trials of convalescent plasma for COVID-19 outside the submitted work. NAG reported receiving salary support from the National Institutes of Health (NIH) National Center for Advancing Translational Sciences via a Johns Hopkins Clinical and Translational Science Award outside the submitted work. YB reported being a site investigator for Janssen outside the submitted work. SMB reported service as chair of a data and safety monitoring board for a Hamilton clinical trial in respiratory failure; fees paid to Intermountain Healthcare from Faron Pharmaceuticals and Sedana Pharmaceuticals for steering committee service for a clinical trial in acute respiratory distress syndrome; research grants to Intermountain Healthcare from Janssen, NIH, Centers for Disease Control and Prevention, and Department of Defense; and royalties from Oxford University Press and Brigham Young University, outside the submitted work. MEC reported service on a Roche advisory board and as a site investigator for Janssen outside the submitted work. This does not alter the authors’ adherence to PLOS ONE policies on sharing data and materials. All other authors report no conflicts of interest.

Response: We note that this sentence was included already as the second-last sentence of our Competing Interests statement, as quoted above: “This does not alter the authors’ adherence to PLOS ONE policies on sharing data and materials.” However, we have changed “the authors’” to “our” and removed the sentence that followed (“All other authors report no conflicts of interest”). We have included the updated Competing Interests statement in our cover letter as requested.

3. We noted in your submission details that a portion of your manuscript may have been presented or published elsewhere. Please clarify whether this publication was peer-reviewed and formally published. If this work was previously peer-reviewed and published, in the cover letter please provide the reason that this work does not constitute dual publication and should be included in the current manuscript.

Response: This work has been posted to the medRxiv preprint server (DOI: 10.1101/2022.01.10.22269008), and aspects of the work were presented at the Methods in Evidence Synthesis Salon at the University of Bristol. As is typical in meta-analyses and as noted in our cover letter, some of the included studies have been published previously. The meta-analysis itself has not otherwise been published or presented.

4. One of the noted authors is a group or consortium [insert name of group or team]. In addition to naming the author group, please list the individual authors and affiliations within this group in the acknowledgments section of your manuscript. Please also indicate clearly a lead author for this group along with a contact email address.’ 

Response: We note that the consortium (Pandemic Response COVID-19 Research Collaboration Platform for HCQ/CQ Pooled Analyses) comprises nonauthor collaborators, as defined by ICMJE. In accordance with ICMJE recommendations and MEDLINE policies (https://www.nlm.nih.gov/bsd/policy/authorship.html), we included the consortium in the byline to ensure the collaborators will be indexed appropriately. We have moved the names and affiliations of the nonauthor collaborators in the consortium from the supplementary appendix to the acknowledgments section of the manuscript. If a contact person for the group is still needed, Anne Gadomski can be contacted at anne.gadomski@bassett.org.

5. We note that you have referenced (ie. Bewick et al. [5]) which has currently not yet been accepted for publication. Please remove this from your References and amend this to state in the body of your manuscript: (ie “Bewick et al. [Unpublished]”) as detailed online in our guide for authors

Response: We have been unable to find “Bewick et al” in our manuscript or our reference list. From your excerpt, it appears to be citation #5, which was Borba et al, published in 2020. We will gladly remove any unpublished citations but are unclear to which you refer.

Reviewer Comments to the Author

Reviewer #1

Thanks for inviting me to reviewing this paper. This is a high-quality IPD meta-analysis. I especially enjoy the statistical methods used for the analysis. Although the topic has already been reported by several other meta-analyses, it still brings value through the individual participant data perspective— the evidence could be ranked at the top over previous similar meta-analyses.

Response: We thank Reviewer #1 for their positive comments on our analysis and its value.

R1: I only have two questions, about the safety assessment. I see the author lists and found there some renewed methodologist in safety assessment. They may have their own consideration. But I think a further investigation may makes the paper more attractive.

1. First, the authors seem failed to define the AEs and SAEs clearly. Ambiguous definition raises confusions during data extraction and impacts the transparency as well as reproducibility.

Response: Thank you for pointing out the potential for ambiguity regarding our definitions of AEs and SAEs. Our AE/SAE definitions were based on each individual protocol’s AE and SAE definitions. We considered harmonizing our AE/SAE definitions across studies but decided against it because this would have required extensive manual chart review by participating trialists. 

We believe that our descriptive analysis remains interpretable and useful to readers because most trialists use standard AE/SAE definitions (most commonly, Common Terminology Criteria for Adverse Events [CTCAE]) so that heterogeneity of definitions across trials is likely to be low.

Notwithstanding this, we have endeavored to clarify this limitation by making the following additions to the manuscript. In the Methods, we added the sentence: “Due to practical constraints, we did not attempt to synchronize adverse event definitions across the included studies.”

In the Discussion, we have added the following limitation: “Sixth, in our safety analysis we did not harmonize AE and SAE definitions among the included studies.”

R1: 2. Second, the authors reported the adverse events outcomes descriptively, this is not the best solution because without weighting scheme the results tend to at risk of Simpson’s paradox. To avoid this, I suggest the authors statistically analysis the safety outcomes, for example, the GLMM/Beta-Binominal model.

Response: We decided not to analyze the safety data using statistical models for the sake of simplicity, particularly given the non-harmonized event definitions, potential differences in safety monitoring and length of follow-up among the studies, and low numbers of events in the smaller studies. Adverse events are commonly presented in an exploratory way in clinical trials, and the case has been made that this is often the appropriate approach (Huster 1991; doi:10.1177/009286159102500315). However, if this explanation is unsatisfactory, we would be happy to conduct a statistical analysis at the editor’s request.

We agree that our pooled results are potentially vulnerable to aggregation bias/Simpson’s paradox. However, we also presented within-study comparisons of safety outcomes, which are not subject to this concern, and our aggregate results are consistent with the within-study from the largest studies. We have added a statement clarifying this to the Results: “These comparisons are potentially vulnerable to aggregation bias, but are broadly consistent with results from within each of the largest three studies (ORCHID, TEACH, and HAHPS).”

Reviewer #2

I read the manuscript carefully. It’s a well-written and robust manuscript only there are some minor comments.

Minor comments:

1. Please add reference for PRISMA guide line.

Response: We thank Reviewer #2 for their helpful comments on ensuring clarity of our use of the PRISMA guidelines. We used the PRISMA extension for individual patient data (PRISMA-IPD). We have updated the manuscript text to clarify this and have added a reference to the PRISMA-IPD statement paper (doi:10.1001/jama.2015.3656). 

The manuscript has been updated in the final sentence of the Methods section as follows: “This study followed the Preferred Reporting Items for Systematic Reviews and Meta-analyses (PRISMA) reporting guideline extension for IPD analyses (PRISMA-IPD) [38].”

R2: 2. the authors claimed that their study stick to PRISMA guide line. However, based on this protocol, search strategy should be presented in text or supplementary file. Also, PRISMA check list is necessary.

Response: Our PRISMA-IPD checklist was included in our original submission as the S3 Appendix. To make it easier to find, we have made it the S1 Appendix so that it is the first item among our supporting information. 

Our search strategy is presented briefly in the Methods section’s Trials Selection Summary, summarized in Fig 1, and outlined in greater detail in our supporting information. Our original manuscript presented the detailed description in S1 Fig: Trial Selection/RCT Selection Process in Detail, a more comprehensive flow diagram with a full page of text afterward that explained our process. We have relabeled that as S3 Appendix: Search Strategy in Detail and placed the text before the figure. Additionally, we have added a new table, Summary of Search Strategies, on the first page of the appendix that presents the search strategy more clearly and accessibly.

Reviewer #3

I carefully read the manuscript by Di Stefano et al. My comments and suggestions for the authors are the following:

1. The search process should be updated (this is a critical issue!). 

Response: We thank Reviewer #3 for their comments and suggestions. We repeated our searches of ClinicalTrials.gov in July 2022. After filtering based on our eligibility criteria (i.e., excluding studies that were international, outpatient, had no enrollment, or were prophylactic), we found two additional potentially eligible studies: NCT04429867 (a randomized trial of hydroxychloroquine versus placebo) and NCT04458948 (a single arm combined study of azithromycin and hydroxychloroquine). Their study listings were last updated June 16 and July 7 of 2020, respectively (our most recent search was June 2, 2020). The first of these, NCT04429867, indicated they had no plans to share IPD.

Though two years between search and publication may be a long interval for traditional aggregate-data meta-analyses, this timeline is typical for IPD meta-analyses. For example, Individual Participant Data Meta-Analysis: A Handbook for Healthcare Research by Riley et al (2021), states: 

“Generally, IPD meta-analysis projects will take upwards of two years to complete, and sometimes longer depending on how many trials are involved and the complexities of negotiating collaboration, data coding, checking, cleaning and analysis” (p. 33) and “IPD meta-analysis projects are time-consuming, [...] often taking upwards of two years to engage with trial investigators; obtain, clean, harmonise and meta-analyse the IPD; and publish and disseminate results” (p. 294).

The Cochrane Handbook for Systematic Reviews of Interventions similarly states: “However, IPD reviews can take longer than other reviews; those evaluating the effects of therapeutic interventions typically taking at least two years to complete.” (Tierney JF, Stewart LA, Clarke M. Chapter 26: Individual participant data. In: Higgins JPT, Thomas J, Chandler J, Cumpston M, Li T, Page MJ, Welch VA [editors]. Cochrane Handbook for Systematic Reviews of Interventions version 6.1 [updated September 2020]. Cochrane, 2020. https://training.cochrane.org/handbook/current/chapter-26)

The literature review in our Discussion section covers more recent studies, including meta-analyses.

We have added the following to the Limitations section of the Discussion to explain this: “Seventh, our final study search was conducted in June 2020. This timeline is consistent with what is typical for IPD meta-analyses, which commonly take upwards of two years to complete [31]. Repeating our searches of ClinicalTrials.gov in July 2022 and applying the same eligibility criteria revealed two unpublished studies (NCT04429867 and NCT04458948) potentially eligible for inclusion, registered in June and July of 2020, respectively. (Registrations of new trials of hydroxychloroquine for COVID-19 had slowed substantially by August 2020 [77].)”

R3: 2. The authors mention that their meta-analysis is PRISMA compliant. However, they should specify the version of PRISMA guidelines they referred to and include the article among the references of the manuscript.

Response: We used the PRISMA extension for individual patient data (PRISMA-IPD). We have updated the manuscript text to clarify this and have added a reference to the PRISMA-IPD statement paper.

The manuscript has been updated in the final sentence of the Methods section as follows: “This study followed the Preferred Reporting Items for Systematic Reviews and Meta-analyses (PRISMA) reporting guideline extension for IPD analyses (PRISMA-IPD) [38].”

Additionally, we have moved our PRISMA-IPD checklist from the S3 Appendix to S1 Appendix so it will be more visible.

R3: 3. Line 212: Including in a meta-analysis data from a personal communication is not a recomendable and increase the risk of publication biases for the analysis.

Response: Our search strategy was accompanied by extensive outreach efforts to the research community designed to encourage broad collaboration and data sharing in COVID studies. We wanted to be transparent about considering the study suggested via personal communication. However, our actual study selection was guided only by our ClinicalTrials.gov searches. As that study was not registered, it was excluded from our analysis.

We have updated the first sentence of our manuscript’s Results section to clarify that exclusion: “Of 19 RCTs identified in our searches (18 from ClinicalTrials.gov; 1 from personal communication that was excluded due to lack of registration on ClinicalTrials.gov), 8 met final criteria for inclusion in our analysis…”

We have addressed this further in our S3 Appendix: Search Strategy in Detail by adding a new sentence and clarifying that the study excluded for lack of registration was the one identified via personal communication: “Although research community outreach was an important component of our process, our study selection was dictated by the ClinicalTrials.gov searches. Of the 19 studies, we selected 8 after excluding 2 that declined participation or did not respond; 3 with ineligible trial designs including outpatient and prophylaxis studies; 2 with no enrollment; 1 not registered on ClinicalTrials.gov (identified via personal communication); and 3 with sites located outside the US (e.g., RECOVERY and DisCoVeRy, part of SOLIDARITY).”

R3: 4. The authors should perform also Begg and Egger's tests.

Response: We have conducted Begg and Egger’s tests. However, it is generally not recommended to use such tests in meta-analyses with fewer than 10 studies (Sterne et al, 2011; doi:10.1136/bmj.d4002) so we have put the p-values themselves in the supporting information. 

We also added results on the estimated between-study standard deviation in treatment effects to facilitate comparison with traditional aggregate-data meta-analyses.

We updated the Results section of the manuscript as follows: “We found no appreciable heterogeneity in estimated treatment-study interactions among the 8 studies after adjusting for individual-level baseline covariates (𝜏 = 0.87 on the log odds scale, 95% CrI, 0.01-5.17); tests for publication bias were inconclusive (S1 Fig).”

We added to the S1 Fig caption the following: “Estimated between-study standard deviations were 1.48 for the “Control” terms (95% CrI 0.16-5.98; on the log-odds scale) and 0.87 for the treatment effects (95% CrI 0.01-5.17). Tests of publication bias applied to unadjusted study-specific proportional ORs from the 6 studies with patients assigned to both HCQ/CQ and control yielded inconclusive results (Egger’s test: p=0.10; Begg’s test: p=0.57); these should be interpreted with caution given the small number of included studies.”

We also renamed the S1 Fig from “Estimated Study Coefficients” to “Assessing Between-Study Heterogeneity.”

R3: The authors should consider to refer to doi: 10.1093/ehjcvp/pvaa105 in the discussion of the manuscript.

Response: Thank you for the suggestion. We reviewed "Management of pregnancy-related hypertensive disorders in patients infected with SARS CoV-2: pharmacological and clinical issues" in consideration for our Discussion but ultimately thought that it was unnecessary to add it to the already extensive section.

Reviewer #4

The authors present a very interesting IPD meta-analysis based on 8 RCTs including unanalyzed data from early trials on the efficacy and safety of HCQ/CQ in COVID-19 prevention and treatment. The paper and analysis are overall of very good quality. I suggested some minors restructuration and additional subgroup analysis that could help strengthen the authors’ claim.

Response: We thank Reviewer #4 for their well-sourced and constructive comments.

R4: Abstract. Authors did not mention what was the primary outcome in the method (they just mentioned the metric of the scale). It is important to know upfront the main measure of their work (at least the conceptual framing behind it). The authors mentioned again the “COVID-19 ordinal scores” in results without giving a specific measure in the abstract. They could conclude on the potential harming effects of HCQ/CQ.

Response: We have clarified in the Abstract that we measured effects using proportional (cumulative) odds ratios by adding the clause: “The primary outcome was a 7-point ordinal scale measured between day 28 and 35 post enrollment; comparisons used proportional odds ratios.”

This is also described in the Methods section.

R4: Introduction. I really appreciate the transparency protocol the authors put into place (though it would be great to have the exact link for Vivli and CCP repositories to be able to consult the database). 

Response: The general links to Vivli (https://www.vivli.org) and the Covid Collaboration Platform (http://covidcp.org) are included in the manuscript in the Data collection and harmonization and Trials selection summary sections, respectively. However, the exact links to the actual datasets are part of our Data Availability Statement. That statement will be included in the paper if accepted but is provided as part of the submission form. It reads:

“The ORCHID trial data underlying the results presented in the study are available from the National Heart, Lung, and Blood Institute Biologic Specimen and Data Repository Information Coordinating Center (https://biolincc.nhlbi.nih.gov/; accession number HLB02372021a). The data for the other studies presented are available from Vivli (https://www.vivli.org): COVID MED, https://doi.org/10.25934/00006535; HAHPS, https://doi.org/10.25934/00006626; NCT04335552, https://doi.org/10.25934/00006861; NCT04344444, https://doi.org/10.25934/00006865; OAHU-COVID19, https://doi.org/10.25934/00006595; TEACH, https://doi.org/10.25934/00006627; and WU352, https://doi.org/10.25934/00006713. Policies for accessing these third-party datasets vary somewhat by study and repository, but requests must be approved and require a signed agreement.” 

R4: Outcomes are better explained in the main text; however, authors could justify why they used these specific levels. Was this recommendation base on the studies they re-analyzed? Or is this just a general classification recommended by health authorities and authors had to recode the whole thing? Precisions may help in that part.

Response: We have added the following sentence to the Outcomes subsection of the Methods explaining our choice: “The primary outcome was clinical improvement measured on a 7-point ordinal scale with levels (1) death; […] (7) not hospitalized, without limitations. This scale is relatively coarse compared with others in use (for example the 11-point WHO scale) [34], and was chosen to make the data easier to harmonize.”

R4: Also, authors mentioned a change in the outcome window due to missing data. How did the authors handle missing data in their study?

Response: We describe the handling of missing covariates in the Statistical Analysis section, but have added a clause to clarify how we handled missing outcome data: "Missing baseline covariates were imputed using multiple imputation by chained equations, as implemented in the R package “mice” (version 3.12); missing outcomes were treated as missing at random conditional on the included covariates."

R4: PLOS One is a very generalist journal so it would be great if the authors could recall the strength-weakness of RCTs vs. observational studies for a general audience as well as the benefits of randomization and how it helps to make causal inferences (i.e., controlling for confounding, random sampling criteria, etc. they could cite Schulz et al., 2002 or Vandenbroucke, 2004 for instance). This could also help strengthen the theoretical part of the paper which is unbalanced compared to the other sections of the paper.

Response: We thank Reviewer #4 for their suggestion that we provide more background for a generalist audience. We agree with this suggestion; however, after considering how best to address the issue, we thought it might be even more helpful to include background on our particular study design (individual participant data meta-analysis) and its advantages over individual trials and more traditional meta-analyses of aggregate data (which should be relatively familiar to readers), rather than a high-level discussion of RCTs versus observational studies.

We expanded the Introduction as follows, including adding a citation per your suggestion to Schulz et al, 2002 (#32 below), on external validity:

“The purpose of this study was to ensure utilization of data from unpublished RCTs evaluating HCQ/CQ by combining them with published data, and to synthesize evidence on HCQ/CQ efficacy and safety in hospitalized COVID-19 patients, overall and in subpopulations of interest, by conducting an individual participant data (IPD) meta-analysis. This study design involves pooling subject-level data from multiple studies and possesses many advantages over both individual randomized trials and traditional aggregate-data meta-analyses. Individual trials are usually designed to detect overall effects; the increased sample size in an IPD meta-analysis can enable more precise estimation of subgroup effects [31]. A more diverse sample in a pooled analysis can also improve external validity over individual trials [32]. Compared with aggregate data meta-analyses, IPD meta-analyses are less vulnerable to the ecological fallacy, allow for consistent analytic choices within each study, and enable researchers to consider subgroup effects that were not considered in the original studies [33].”

R4: Results. The authors mentioned early on that, due to missing data, they had to broaden days of post enrollment from 28-30 to 28-35. It would still be interesting though to check whether the main results changed in function of this amendment and how the authors can justify days 35 as the cut-off and not 34 or 33. 

Response: The manuscript references sensitivity analyses we conducted using both expanded (day 28-40) and contracted (day 28-30) outcome windows. These did not result in qualitatively different conclusions.

We have added the following paragraph to the Results explaining the choice of day 28-35 window:

“Data on the prespecified primary outcome (the 7-point ordinal scale measured between days 28 and 30) was available for 90% of patients (695 out of 770). In one study, however (TEACH) data was available for only 45% of patients (58 out of 128). Because of this, the decision was made to broaden the primary outcome window to days 28-35. This decision was made without examining the outcome data themselves. With the broader definition, primary outcome data was available for 76% of patients in TEACH (97 out of 128) and 95% of patients overall (734 out of 770).”

We have amended the Outcomes section of the Methods to refer to this:

“We prespecified an outcome window of day 28-30 post-enrollment, which was broadened to day 28-35 after data collection due to missingness (see Results).”

R4: The authors make a compelling case of justifying many sub-group analyses to explore whether the HCQ/CQ presented local benefits for age, gender, BMI, etc. subgroups. Another possible and important subgroup is the timing of the medication. Some authors (see Prodomos & Rumschlag, 2020 or Million et al., 2020) have argued that HCQ is only effective when provided earlier, not associated with worsening disease and safe. Although I’m aware this argument is very borderline, if the authors have enough studies to compared early vs. late treatment, they would be able to respond to those critics and relativize some open-access analysis that claim early treatment showed better improvement.

Response: We have conducted a post-hoc, plug-in subgroup analysis of days between symptom onset and enrollment. We considered adding days between symptom onset and enrollment to our main statistical model, but decided against this on the grounds that it would represent a large departure from pre-registration.

In the supporting information we have added the new S5 Fig and text: 

“Post-hoc exploratory analysis of subgroups based on time between symptom onset and enrollment. Subgroups are based on approximate tertiles. Shown are (A) proportional odds ratios from models fit by maximum likelihood within each subgroup, with 95% confidence intervals; (B) the empirical risk of survival for each treatment group within each subgroup, with 95% confidence intervals; and (C) empirical risk differences within each subgroup, with 95% confidence intervals.”

We also made the following changes to the manuscript. 

In the Methods section: “We conducted two post-hoc subgroup analyses. The first used quintiles of a baseline risk score given by the expected linear predictor for each study participant under the control condition, as per recommendations from Kent et al [35]. The second was based on the time between symptom onset and enrollment (0-4 days, 5-7 days, ≥8 days; groups based on approximate tertiles).

In the Results section:“There were no substantial effects of HCQ/CQ observed in any prespecified subgroup, nor in a post-hoc subgroup analysis based on the time between symptom onset and enrollment (Fig 2; S5 Fig).”

We have also added this as a post-hoc SAP modification in S3 Table.

R4: Discussion. The discussion part is much more detailed than the introduction. The authors provide a review of RCTs history on HCQ and COVID-19. I would recommend moving some parts of the discussion to the introduction to better contextualize the debate on HCQ and help naive readers understand what’s at stake and why their work is important.

The Annex and online supplementary materials are very complete and detailed.

Response: We considered this suggestion carefully and decided against it for two reasons. First, our discussion of this history of HCQ for COVID-19 is arranged chronologically, and we were concerned that splitting it up might reduce clarity and be harder for readers to follow. Second, our preference is for a simpler Introduction to get readers to our own study’s contributions relatively quickly, with a more in-depth context given in the Discussion after we have presented our main results. However, we welcome the editor’s thoughts and would be happy to reconsider this if he agrees it would improve the manuscript.

---

## [Decision Letter · Decision Letter 1]

10 Aug 2022

Hydroxychloroquine/Chloroquine for the Treatment of Hospitalized Patients with COVID-19: An Individual Participant Data Meta-Analysis

PONE-D-22-06839R1

Dear Dr. Freilich,

We’re pleased to inform you that your manuscript has been judged scientifically suitable for publication and will be formally accepted for publication once it meets all outstanding technical requirements.

Kind regards,

A. M. Abd El-Aty

Academic Editor

PLOS ONE

Additional Editor Comments (optional):

Reviewers' comments:

Reviewer's Responses to Questions

**Comments to the Author**

1. If the authors have adequately addressed your comments raised in a previous round of review and you feel that this manuscript is now acceptable for publication, you may indicate that here to bypass the “Comments to the Author” section, enter your conflict of interest statement in the “Confidential to Editor” section, and submit your "Accept" recommendation.

Reviewer #1: All comments have been addressed

Reviewer #3: All comments have been addressed

Reviewer #4: All comments have been addressed

2. Is the manuscript technically sound, and do the data support the conclusions?

Reviewer #1: Yes

Reviewer #3: Yes

Reviewer #4: Yes

3. Has the statistical analysis been performed appropriately and rigorously? 

Reviewer #1: Yes

Reviewer #3: Yes

Reviewer #4: Yes

4. Have the authors made all data underlying the findings in their manuscript fully available?

Reviewer #1: Yes

Reviewer #3: No

Reviewer #4: Yes

5. Is the manuscript presented in an intelligible fashion and written in standard English?

Reviewer #1: Yes

Reviewer #3: Yes

Reviewer #4: Yes

6. Review Comments to the Author

Reviewer #1: (No Response)

Reviewer #3: (No Response)

Reviewer #4: The manuscript is technically sound and the analysis have been performed correctly. The authors intregrated all my comments and answered relevantly to those which they disagreed with.

7. PLOS authors have the option to publish the peer review history of their article (what does this mean?). If published, this will include your full peer review and any attached files.

Reviewer #1: **Yes: **Chang Xu

Reviewer #3: No

Reviewer #4: **Yes: **Jordane Boudesseul

---

## [Editor Report · Acceptance letter]

18 Aug 2022

PONE-D-22-06839R1 

Hydroxychloroquine/chloroquine for the treatment of hospitalized patients with COVID-19: An individual participant data meta-analysis 

Dear Dr. Freilich:

I'm pleased to inform you that your manuscript has been deemed suitable for publication in PLOS ONE. Congratulations! Your manuscript is now with our production department. 

Kind regards, 

on behalf of

Prof. A. M. Abd El-Aty 

Academic Editor

PLOS ONE